# Targeting the transmembrane cytokine co-receptor neuropilin-1 in distal tubules improves renal injury and fibrosis

Yinzheng Li[1], Zheng Wang[1], Huzi Xu[1], Yu Hong[1], Mengxia Shi[1], Bin Hu[1], Xiuru Wang[1], Shulin Ma[1], Meng Wang[1], Chujin Cao[1], Han Zhu[1], Danni Hu[1], Chang Xu[1], Yanping Lin[1], Gang Xu[1] ✉, Ying Yao [1,2] ✉ & Rui Zeng [1,3] ✉

Neuropilin-1 (NRP1), a co-receptor for various cytokines, including TGF-β, has been identified as a potential therapeutic target for fibrosis. However, its role and mechanism in renal fibrosis remains elusive. Here, we show that NRP1 is upregulated in distal tubular (DT) cells of patients with transplant renal insufficiency and mice with renal ischemia-reperfusion (I-R) injury. Knockout of *Nrp1* reduces multiple endpoints of renal injury and fibrosis. We find that Nrp1 facilitates the binding of TNF-α to its receptor in DT cells after renal injury. This signaling results in a downregulation of lysine crotonylation of the metabolic enzyme Cox4i1, decreases cellular energetics and exacerbation of renal injury. Furthermore, by single-cell RNA-sequencing we find that *Nrp1*-positive DT cells secrete collagen and communicate with myofibroblasts, exacerbating acute kidney injury (AKI)-induced renal fibrosis by activating Smad3. Dual genetic deletion of *Nrp1* and *Tgfbr1* in DT cells better improves renal injury and fibrosis than either single knockout. Together, these results reveal that targeting of NRP1 represents a promising strategy for the treatment of AKI and subsequent chronic kidney disease.

Many studies have shown that elevated expression or activity of transforming growth factor-beta (TGF-β) contributes to progressive human kidney diseases, and animal models have identified TGF-β as a key pathogenic driver of organ fibrosis, including renal tubulointerstitial fibrosis[1,2]. Fibrosis is characterized by upregulation of the synthesis of matrix proteins, inhibition of matrix degradation, and modulation of cell-cell interactions during pathological organ regeneration and repair[3]. Currently, several therapeutic strategies targeting TGF-β have been investigated, including interventions aimed at inhibiting its production, activation, receptor binding, and intracellular signaling[3]. Although some of these strategies have shown promise in clinical trials against renal fibrosis, their overall clinical efficacy remains limited[4,5]. Additionally, small molecule inhibitors targeting

TGF-β ligands have exhibited pre-clinical dose-limiting toxicities, resulting in cardiac valve lesions[6] and aortic aneurysms[7]. Therefore, exploring new therapeutic approaches that effectively target TGF-β signaling while reducing adverse events is imperative and has important clinical significance.

Current TGF-β inhibitors primarily target TGF-β and its receptors. However, TGF-β escape[8] or TGF-β-independent SMAD 2/3 activation[9] may seriously reduce the efficacy of these TGF-β inhibitors. In addition to the direct ligand-receptor combinations, the potency of TGF-β signaling is also mediated by cell surface co-receptors[10]. Unlike classical TGF-β receptors, co-receptors do not possess functional kinase domains and instead have short cytoplasmic domains that initiate more abundant cell surface ligand

[1]Division of Nephrology, Tongji Hospital, Tongji Medical College, Huazhong University of Science and Technology, 1095 Jiefang Ave, Wuhan 430030, China. [2]Department of Nutrition, Tongji Hospital, Tongji Medical College, Huazhong University of Science and Technology, 1095 Jiefang Ave, Wuhan 430030, China. [3]Key Laboratory of Organ Transplantation, Ministry of Education, NHC Key Laboratory of Organ Transplantation, Key Laboratory of Organ Transplantation, Chinese Academy of Medical Sciences, Wuhan 430030, China. ✉e-mail: xugang@tjh.tjmu.edu.cn; yaoyingkk@126.com; zengrui@tjh.tjmu.edu.cn

interaction[10]. Co-receptors may also interact with classical TGF-β receptors to influence the recruitment of receptor complex components, alter their stability, and influence intracellular trafficking patterns[10]. Thus, the benefit of targeting TGF-β co-receptors is the potential for treatments directly targeting core components of these classical TGF-β pathways[10].

Notably, neuropilin-1 (NRP1), a broad-spectrum cytokine co-receptor, including for TGF-β, plays a crucial role in augmenting TGF-β signaling[11]. NRP1 is classified as a cell surface glycoprotein of the non-tyrosine kinase type. It functions as a single-pass transmembrane receptor protein, lacking enzymatic activity, and possesses an extensive extracellular tail with multiple domains[12,13]. This structural feature enables NRP1 to interact with various ligands participating in diverse signaling pathways. These ligands include class III/IV semaphorins, vascular endothelial growth factors and TGF-β[14,15]. Many studies have shown that NRP1 has strong profibrotic properties[16–20]. Upregulation of NRP1 in hepatic stellate cells and mesenchymal cells leads to excessive activation of platelet-derived growth factor receptor-beta (PDGFRβ) signaling, driving matrix expansion[16,17], while specific deletion of *Nrp1* in hepatic stellate cells ameliorated liver inflammation and fibrosis[18]. Moreover, NRP1 promotes abnormal growth factor signaling in breast cancer, resulting in EMT-related drug resistance and cancer metastasis[19]. Thus, NRP1 is a potential therapeutic target for liver fibrosis and pulmonary fibrosis[20], as well as cancer. However, as a co-receptor of TGFR1, the precise relationship between NRP1 and renal fibrosis remains elusive.

In addition to TGF-β, NRP1 has also been found to interact with TNF-α[21,22]. The upregulation of NRP1 in the rat middle cerebral artery occlusion/ischemia-reperfusion (MCAO/IR) model results in activation of the NF-κB signaling pathway, exacerbating neuronal inflammation and aggravating brain ischemia-reperfusion injury[23]. However, the role of NRP1 as a co-receptor of TNF-α in renal tubular epithelial cells remains to be elucidated.

A mouse model of ischemia-reperfusion (I-R)-induced kidney injury is widely utilized to replicate human acute kidney injury (AKI). It is well-established that I-R induces a high degree of damage to proximal tubules; therefore, numerous previous studies have predominantly concentrated on investigating this particular cell type[24,25]. For instance, Yoshiharu Muto et al. recently demonstrated that increased chromatin accessibility of NF-κB binding sites in failed-repair proximal tubules (PT) exacerbate tubular injury, and promote inflammation and fibrosis following I-R-induced injury[26].

Nevertheless, recent research has revealed that, apart from proximal tubules, there are substantial transcriptomic responses to AKI in thick ascending limbs (TALs) and distal convoluted tubules (DCTs). Furthermore, key renal injury markers, like KIM1 and TIMP2, are notably enriched in these cell types[27], indicating damage to distal tubules (DTs). A recent study showed that although normal gene is restored in proximal tubules during advanced kidney fibrosis, distal segments of the nephron tubule endure persistent and unresolved injury[28], suggesting DTs plays a key role in the AKI-to-chronic kidney disease (CKD) transition.

Here, we utilized a mouse model of bilateral I-R to induce renal injury and subsequent renal fibrosis. We found that *Nrp1* expression was upregulated in distal and injured tubule cells after I-R-mediated injury. Knockout of *Nrp1* in renal DT epithelial cells ameliorates kidney injury and subsequent chronic renal fibrosis. Our findings indicated that Nrp1 regulates the TNF-α-dependent expression of Acox3, a crucial enzyme involved in the regulation of the key electron transport chain enzyme Cox4i1 in cellular oxidative phosphorylation (OXPHOS). We also found that Nrp1 is involved in the progression of renal disease by activating TGF-β-dependent and -independent Smad 2/3 signaling in distal renal tubular epithelial cells (TECs). Our findings provide insights into a potential therapeutic approach for improving kidney disease by targeting NRP1.

## Results

### Nrp1 expression increases in distal TECs after I-R-induced renal injury

To assess the activation of Nrp1 and Tgf receptors in kidney injury following I-R, we conducted single-cell RNA-sequencing (scRNA-seq) on kidney samples from mice at 4 h, 12 h, 1 day, 5 days, 14 days and 28 days after I-R (Supplementary Fig. 1A). A total of 17 distinct cell clusters were identified (Supplementary Fig. 1B-1C). Analysis of the scRNA-seq data revealed notably higher *Tgfbr1* expression in fibroblasts and myofibroblasts, as well as in DT cells, after I-R injury compared to non-injured cells (Fig. 1A). *Tgfbr2* expression was mainly higher in principal cells of the collecting duct, myofibroblasts, macrophages and pDCs, while *Tgfbr3* expression was higher in myofibroblasts and pericytes compared to cells in sham group (Supplementary Fig. 1D). After I-R-induced injury, *Nrp1* expression in distal and injured tubule cells was higher than in the sham control (Fig. 1B), which was confirmed by data from the public kidney single cell datasets at http://humphreyslab.com/SingleCell/ (Fig. 1C)[29]. By fluorescence in situ hybridization (FISH), we found co-staining of a *Nrp1* probe with protein expression of Cdh16, a specific marker for the distal portion of the nephron[30], confirming the expression of *Nrp1* in the distal renal tubules (Fig. 1D). This finding was further confirmed by data from publicly available kidney single-cell or single-nuclei datasets from https://singlecell.broadinstitute.org/[31,32] and GSE197266 (ref. 33), and the public renal space transcriptomics at https://www.spatialomics.org/SpatialDB/[34] (Supplementary Fig. 2A–D). We further confirmed Nrp1 expression in renal tubular epithelial cells by using antibodies from different commercial vendors, including Santa Cruz, R&D and Abcam (Supplementary Fig. 2E). We found that there was a gradual increase of Nrp1 expression in renal tubule cells following I-R, with peak expression observed at day 5 after I-R (Fig. 1E). Furthermore, Nrp1 expression was higher in kidneys from mice with unilateral ureteral obstruction (UUO) and those with 5/6 nephrectomy than in their respective sham controls (Fig. 1E). We also observed co-expression of Nrp1 with the renal tubular injury marker Kim1 (Supplementary Fig. 2F), while *Nrp2* expression was primarily lower in myofibroblasts and higher in endothelial cells (Supplementary Fig. 2G).

A total of 4 distinct DT cell clusters were identified (Fig. 1F). Co-expression analysis of NRP1 with TGF-β receptors revealed the highest co-expression proportion with TGF-β receptor 1 in DT cells, particularly in the DCTs and loop of Henle (LOH) (Fig. 1G, H and Supplementary Fig. 2H). As it is well known that Nrp1 is widely expressed in glomeruli and peritubular capillaries, we also analyzed the distribution of NRP1 in the renal vasculature and found a downregulation of Nrp1 in endothelial cells following renal injury (Fig. 1B). Meanwhile, considering that TECs are the most widely distributed and directly injured parenchymal effector cells in response to ischemia in the kidney, we focused our subsequent experiments and analyzes on DT cells.

To explore protein interactions, we utilized the STRING database (https://string-db.org/) and identified interactions of NRP1 with both TGF-β and TNF-α (Fig. 1I). In *Nrp1*-expressing DT cells (*Nrp1* + DT), the expression of TNF receptor *Tnfr1* (TNF-α receptor) was significantly higher than in the sham control (Fig. 1J). By co-immunoprecipitation (co-IP) we further confirmed the interaction between Nrp1 and Tnr1a (Fig. 1K). Co-expression analysis of *Nrp1* with TNF-α receptors demonstrated the highest co-expression proportion with *Tnfr1* (Fig. 1L). Immunofluorescence staining confirmed the co-localization of Nrp1 with the renal DT marker S12a3 (Fig. 1M). Simultaneously, Nrp1 co-localized with Tgfr1 and Tnr1a, suggesting that Nrp1 served as a co-receptor for Tgfr1 and Tnr1a (Fig. 1M). These findings indicate that, apart from its potential effects through TGF-β, Nrp1 contributes to kidney injury through the activation of TNF-α signaling.

During kidney transplantation, the process of removing the kidney from the donor and implanting it into the recipient results in I-R, which closely resembles renal I-R in mice. Here, we obtained kidney

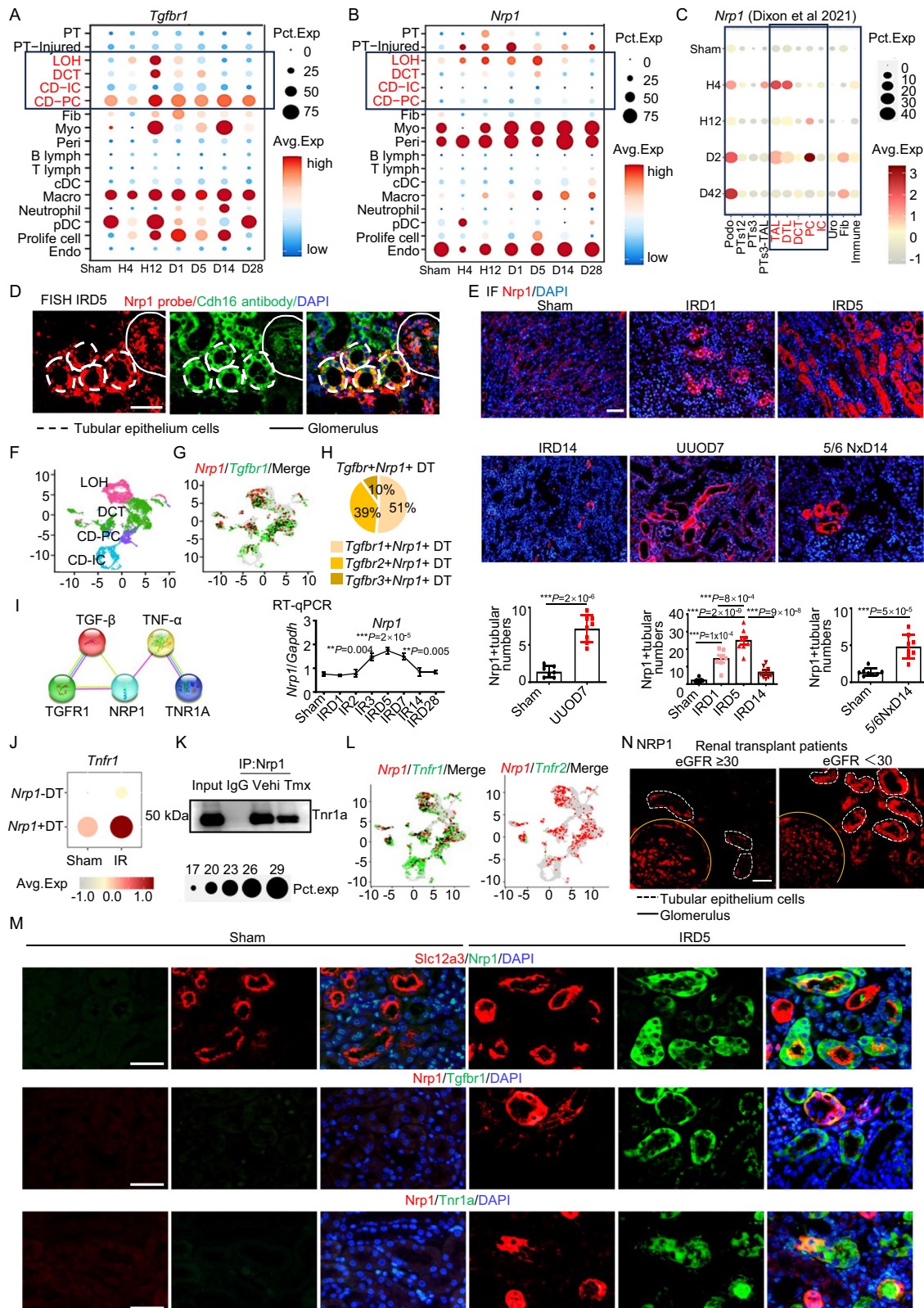

biopsy samples from 217 patients who experienced renal dysfunction after kidney transplantation at Tongji Hospital in Wuhan, China from 2019 to 2023. We found higher NRP1 expression in renal tubular cells in the biopsy samples of patients with a glomerular filtration rate (GFR) < 30 ml/min/1.73 m² with a significant decline in renal function after kidney transplantation than that of patients with a GFR ≥ 30 ml/min/1.73 m² (Fig. 1N). We categorized the renal transplant biopsy

samples into two groups based on the median value of the number of NRP1-positive tubules. Samples with a value greater than or equal to the median were classified as "NRP1-high," while those below the median were classified as "NRP1-low." To investigate the relationship between NRP1 expression and the estimated GFR (eGFR), we analyzed medical records of these 217 patients, including data on gender, age, history of hypertension and diabetes, blood routine,

**Fig. 1 | Nrp1 expression is upregulated in distal TECs of patients with transplant renal insufficiency and mice with IR-induced AKI and CKD. A** Expression levels of *Tgfbr1* in kidney after IR surgery. **B** Expression levels of *Nrp1* in kidney after IR surgery. **C** Expression levels of *Nrp1* in kidney after IR surgery. Figure created with humphreyslab.com. **D** Co-staining Nrp1 and Cdh16 using fluorescence in situ hybridization (FISH). **E** Representative immunofluorescence images of Nrp1 in IR(For Sham, *n* = 7; IRD1, *n* = 7; IRD5, *n* = 8; IRD14, *n* = 8), UUO (For Sham, *n* = 7; UUOD7, *n* = 8), and 5/6 nephrectomy (*n* = 8 per group) mice. **F** UMAP plot displayed the distribution of distal tubular (DT) cells, including loop of Henle (LOH), distal convoluted tubule(DCT), principal cells of collecting duct (CD-PC), and intercalated

cells of collecting duct (CD-IC). **G, H** Co-expression of Nrp1 and TGF-β receptors in DT cells. **I** Interaction relationships between NRP1, TGF-β and TNF-α, and their receptors using the STRING website (https://string-db.org/). **J** Expression level of *Tnfr1* in DT cells. **K** Immunoprecipitation experiments of the interaction between Nrp1 and Tnr1a. **L** Co-expression of *Nrp1* and TNF-α receptors in DT cells. **M** Co-expression of Nrp1 with Tgfr1, Tnr1a and distal tubular marker S12a3 with immunofluorescence staining. The experiments were independently repeated three times. **N** Immunofluorescence staining of NRP1 in kidney transplant patients. *\*P* < 0.05, *\*\*P* < 0.01, *\*\*\*P* < 0.001 as determined by one-way ANOVA. Scale bar, 20 μm. Data represent mean ± SEM. Source data are provided as a Source Data file.

**Table 1 | Risk factors associated with renal function decline in renal transplant dysfunction patients**

| Variable | Univariate analysis | | | Multivariate analysis | | |
|---|---|---|---|---|---|---|
| | OR | 95%CI | *P* | OR | 95%CI | *P* |
| Hypertension (with or without) | 1.626 | 0.783-3.375 | 0.192 | - | - | - |
| Diabetes (with or without) | 0.875 | 0.362-2.113 | 0.134 | - | - | - |
| Lymphocyte,10^9/L | 0.964 | 0.845-1.099 | 0.580 | - | - | - |
| Ca²⁺, mmol/L | 1.037 | 0.92-1.168 | 0.556 | - | - | - |
| TP, g/L | 0.933 | 0.895-0.972 | **0.001** | 1.022 | 0.972-1.075 | 0.386 |
| ALB, g/L | 0.911 | 0.863-0.962 | **0.001** | 0.941 | 0.865-1.023 | 0.151 |
| NRP1 in TECs (high vs low)ᵃ | 2.116 | 1.228-3.646 | **0.007** | 2.536 | 1.376-4.673 | **0.030** |

The bold values indicate *P* < 0.05.
ᵃNRP1 (NRP1 high was defined as the number of NRP1 positive cells ≥50th percentiles.
1, NRP1 positive cells ≥50th percentiles; 0, NRP1 positive cells <50th percentiles).
The statistical analyzes were two-sided.

blood biochemistry, urine routine and the eGFR (Table S1). Based on the eGFR, we divided patients into two groups: those with a severe decline in eGFR (<30 ml/min/1.73 m², *n* = 115) and those with a mild-to-moderate decline in eGFR (≥ 30 ml/min/1.73 m², *n* = 102). Comparisons between the two groups showed statistically significant differences in lymphocyte count, blood calcium level, total protein level, albumin level and the percentage of NRP1-high tubules (*P* < 0.05). Univariate and multivariate regression analyzes revealed that NRP1 serves as an independent risk factor for renal function decline (Table 1).

In summary, our findings demonstrate the upregulation of Nrp1 in distal tubules in diseased kidneys. The interaction of Nrp1 with TGF-β and TNF-α suggests its potential involvement in acute injury and the chronic progression of kidney disease.

## Nrp1 in distal TECs worsens renal injury and renal fibrosis

To investigate the impact of distal tubular Nrp1 on kidney injury and fibrosis, we generated mice with a specific knockout of *Nrp1* in TECs (Nrp1^flox/+^;Ksp-iCre) (Supplementary Fig. 3A). After induction of the gene knockout, we confirmed the expected genotypes of the mice (Supplementary Fig. 3B, C). The gene knockout did not result in significant pathological changes in the heart, liver, spleen, lungs and kidneys (Supplementary Fig. 3D). By RT-qPCR and Western blotting we confirmed the reduced expression of *Nrp1*/Nrp1 after *Nrp1* knockout (Fig. 2A, and Supplementary Fig. 3E). Using these gene knockout mice, we established the presence of I-R-mediated injuries at day 5 (IRD5) and day 14 (IRD14) after induction of I-R (Fig. 2B). On IRD5, compared to I-R groups given vehicle, *Nrp1* knockout in distal tubules not only resulted in lower ratios of kidney-to-body weights, as well as levels of blood urea nitrogen (BUN) and creatinine (CR) (Fig. 2C), but also resulted in less pathological damage, such as necrosis, tubular dilation, tubular cast formation and loss of brush border (Fig. 2D). The renal tubular injury marker Kim1 was also lower in the knockout mice compared to the vehicle-treated controls (Fig. 2D). Furthermore, RT-qPCR confirmed that *Nrp1* knockout alleviated kidney damage markers (*Havcr1* and *Lcn2*) (Fig. 2F). Such improvements in the knockout mice were also seen at IRD14 (Fig. 2C–F), as well as kidney fibrosis

characterized by interstitial collagen deposition (Fig. 2E) and fibrosis-related factors (*Acta2*, *Pdgfrb*, *Col1a1* and *Fn1*) after I-R (Fig. 2F), except that the kidney weight-to-bodyweight ratios in the knockout mice were higher than the vehicle-treated mice, though comparable to the sham controls (Fig. 2C). The changes in kidney weight-to-bodyweight ratios at IRD5 and IRD14 suggested the knockout of *Nrp1* in distal tubules improved inflammatory infiltration and atrophy in the kidney compared to the vehicle-treated controls.

Additionally, we utilized subcapsular injection of *Nrp1*-over-expressing lentivirus to overexpress Nrp1 on the membrane of TECs, as well as GFP intracellularly, and found that such overexpression resulted in exacerbated kidney injury at IRD5 (Supplementary Fig. 3F–K). On IRD1, compared to I-R groups given vehicle, *Nrp1* knockout in distal tubules not only resulted in less pathological damage, but also resulted in lower levels of BUN and CR (Supplementary Fig. 3L, M). The protective effect of *Nrp1* knockout was also observed in kidneys injured by UUO and 5/6 nephrectomy (Supplementary Fig. 4).

In summary, these findings demonstrate that Nrp1 in distal tubules promotes kidney injury and subsequent chronic kidney fibrosis.

## Nrp1 inhibits aerobic metabolism in distal TECs by suppressing crotonylation of glucose metabolic enzymes

The mechanism by which the TNF-α-related pathway in TECs promotes renal injury and the subsequent fibrosis is still controversial[35–37]. To further understand the mechanism by which the NRP1-TNF-α receptor axis promotes renal injury, we analyzed transcription factor activities in *Nrp1* + DT and *Nrp1*-DT. We found that the activities of Nfkb1 and Smad3 were elevated in Nrp1+DT (Fig. 3A). Western blotting analysis confirmed that the expression of Nfkb1, Smad3 and fibrosis-related markers (α-SMA and Pdgfrb) were higher after I-R compared to base line, while knockout of *Nrp1* resulted in lower expression (Fig. 3B). By scRNA-seq analysis we found that in the DT, *Nfkb1* and *Smad3* were expressed at the highest levels in Nrp1+Tgfr1−Tnr1a+DT cells and Nrp1+Tgfr1+Tnr1a−DT cells, respectively (Fig. 3C). We visualized the distribution of *Nrp1* and the key transcription factor activity in DT cells

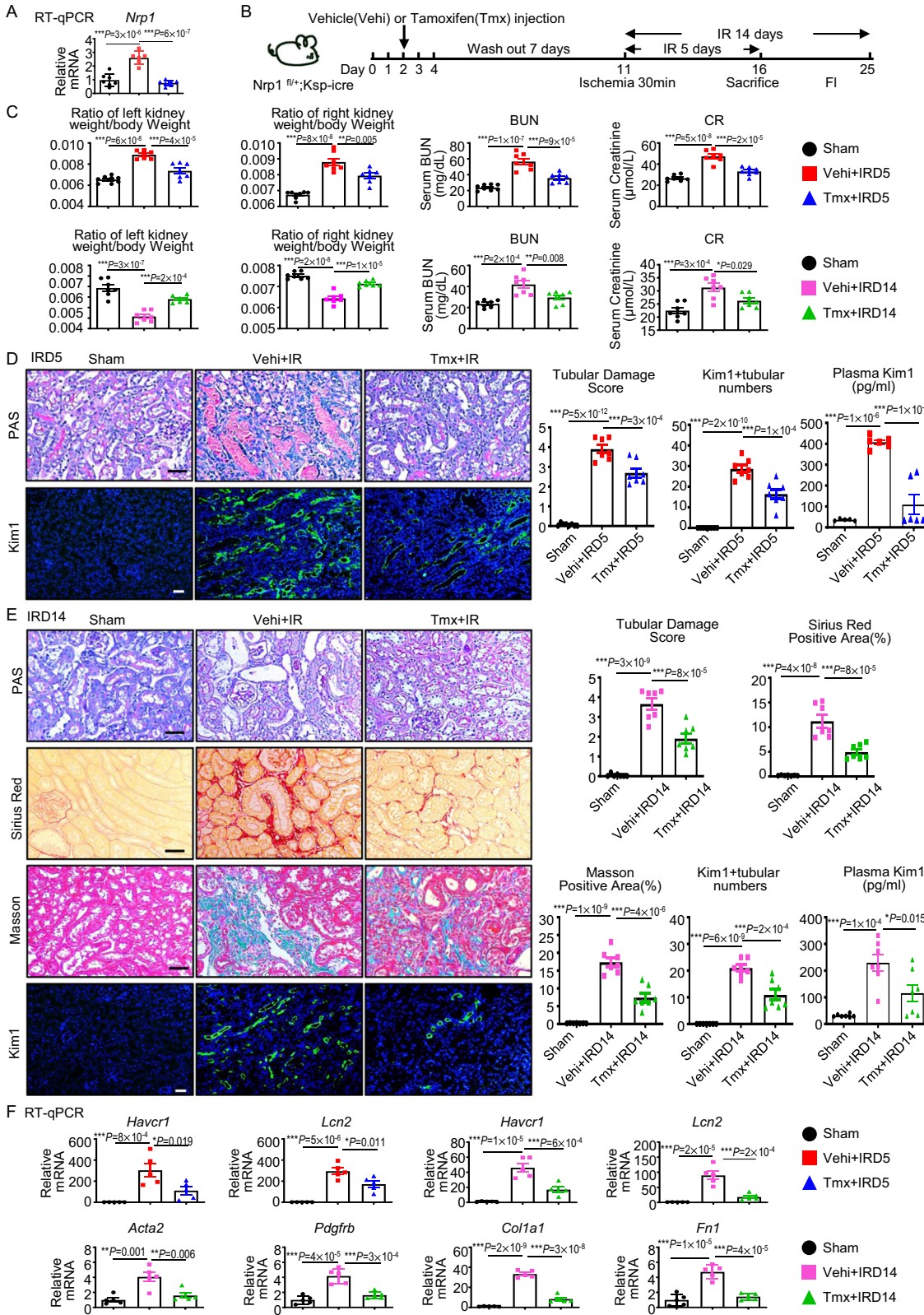

by analyzing the scRNA-seq data (Fig. 3D). We observed that Nfkb1 exhibited higher transcriptional activity in *Nrp1*+ DT cells, whereas Etv6, a transcription factor known mainly for its role in hematology and blood cancers, showed higher activity in *Nrp1*-DT cells (Fig. 3D). To further validate this observation, we performed CUT&RUN experiments and confirmed through qPCR that the addition of an anti-Nfkb1 monoclonal antibody to the lysates of pTECs significantly amplified the

expression of *Etv6*, indicating an increased binding of Nfkb1 to the promoter region of *Etv6* (Fig. 3E) in NRP1-positive cells. These findings suggest that NRP1 enhances the interaction of Nfkb1 with *Etv6* and inhibits the transcription of *Etv6*.

Next, the transcription factor analysis showed that Etv6 positively regulates the expression of *Acox3* (Fig. 3A). Given that Acox3 is an enzyme involved in lysine crotonylation modification (Kcr), we

**Fig. 2 | Knockout of *Nrp1* in TECs reduces I-R-induced kidney damage and fibrosis. A** Levels of *Nrp1* measured by RT-qPCR (For Sham, $n = 7$; Vehi+IRD5, $n = 6$; Tmx+IRD5, $n = 7$). **B** A schematic diagram illustrating the experimental scheme of I-R analysis. **C** Ratio of kidney weight versus body weight of mice. Plasma blood urea nitrogen (BUN) concentrations and creatinine (CR) concentrations in sham, Vehi + IR, or Tmx + IR at 5 days (For Sham, $n = 8$; Vehi+IRD5, $n = 7$; Tmx+IRD5, $n = 7$) and 14 days ($n = 7$ per group). **D** Representative micrographs and corresponding statistical scores of periodic acid-Schiff (PAS) (For Sham, $n = 8$; Vehi+IRD5, $n = 7$; Tmx +IRD5, $n = 7$), Kim1 immunofluorescence staining (For Sham, $n = 8$; Vehi+IRD5, $n = 7$; Tmx+IRD5, $n = 7$) and plasma Kim1 detected by enzyme linked immunosorbent assay (ELISA) (For Sham, $n = 5$; Vehi+IRD5, $n = 6$; Tmx+IRD5, $n = 6$) on day 5 after IR in mice. The assessment of renal tubular damage involves evaluating tubular

necrosis, cast formation, tubular dilation, and brush border loss. Scores are assigned to indicate the degree of damage: 0 for no damage, 1 for 10% damage, 2 for 11–25% damage, 3 for 26–45% damage, 4 for 46–75% damage, and 5 for more than 76% damage. **E** Representative micrographs and corresponding statistical scores of PAS, Masson, Sirius red, Kim1 immunofluorescence staining and plasma Kim1 detected by ELISA on day 14 after IR in mice ($n = 7$ per group). **F** Expression levels of kidney damage indicators (*Havcr1* and *Lcn2*) and fibrosis-related factors (*Acta2*, *Pdgfrb*, *Col1a1* and *Fn1*) at day 14 after IR determined using RT-qPCR ($n = 5$ per group). *$P < 0.05$, **$P < 0.01$, ***$P < 0.001$ as determined by one-way ANOVA. Scale bar, 20 µm. Data represent mean ± SEM. Source data are provided as a Source Data file.

hypothesized a potential association between Etv6 and Kcr. Further analysis of the enzyme-encoding genes involved in the generation of crotonyl-CoA, the precursor of crotonylation modification, revealed that their expression was lower in the early stage of I-R but recovered in the late stage (Fig. 3F). Conversely, decrotonylases and crotonyltransferases that regulate non-histone Kcr showed greater expression in the early stage of I-R and lower expression in the late stage (Supplementary Fig. 5A).

Crotonyl-CoA is a cofactor produced during fatty acid oxidation (FAO) and lysine/tryptophan metabolism, and it plays a key role in the process of Kcr[38]. Enzymes involved in fatty acid and amino acid metabolism that participate in crotonyl-CoA generation, such as acyl-CoA oxidases (ACOX3, ACOX1), short-chain acyl-CoA dehydrogenase (ACADS), medium-chain acyl-CoA dehydrogenase (GCDH) and enoyl-CoA hydratase (ECHS1), increase the levels of Kcr[38] (Fig. 3G). To confirm the changes in Kcr and other classical post-translational modifications after I-R, we performed Western blotting analysis (Fig. 3H and Supplementary Fig. 5B). We observed lower levels of Kcr in the early stage of I-R, followed by higher levels in the late stage after I-R (Fig. 3H, left). However, *Nrp1* knockout led to an early recovery of Kcr to the sham level (Fig. 3H, right). Similar results were observed by scRNA-seq analysis of *Acox3* in DT cells (Supplementary Fig. 5C).

Modification proteomics of Kcr showed a decrease in the level of Kcr in enzymes involved in the tricarboxylic acid cycle (TCA) and in OXPHOS after I-R compared to sham controls, while knockout of *Nrp1* increased their Kcr levels (Fig. 3I). This suggests that Nrp1 influences aerobic metabolism by modulating the levels of Kcr.

Consistent with previous studies, our scRNA-seq analysis demonstrated a significant overall reduction in OXPHOS levels (Supplementary Fig. 6A). Meanwhile, we found a significant reduction in OXPHOS and TCA-related gene expression in *Nrp1* + DT cells compared to *Nrp1*-DT cells (Fig. 3J and Supplementary Fig. 6B–D). By RT-qPCR we found that at IRD5 knockout of *Nrp1* resulted in the restoration of OXPHOS and TCA levels (Supplementary Fig. 7). Dot plots showed reduced mitochondrial function-related gene expression in *Nrp1* + DT cells compared to *Nrp1*-DT cells (Fig. 3K). Mitochondrial staining with MitoTracker in pTECs suggested that *Nrp1* knockout restored mitochondrial function (Fig. 3L). Furthermore, by Seahorse assays of pTECs to measure the levels of OXPHOS and glycolysis, we found that after oxygen glucose deprivation/re-oxygenation (OGD/R), *Nrp1* knockout resulted in a shift from a quiescent state with a low oxygen consumption rate (OCR) and extracellular acidification rate (ECAR) to an energetic state with high OCR and ECAR, as compared to the condition of OGD/R alone (Fig. 3M). These findings confirmed that Nrp1 inhibits aerobic metabolism in distal TECs.

To investigate the detailed role of Nrp1 in aerobic metabolism of distal TECs, we analyzed the proteomic data of Kcr and identified the proteins with the most significant changes in this modification after I-R and *Nrp1* knockout. The lysine crotonylation modification of Cox4i1, Immt and Mdh1 were specifically greater at the sites of K29, K312 and K248, respectively, after *Nrp1* knockout in DT cells (Fig. 4A). By scRNA-seq analysis we found that the gene expression of *Immt*, *Mdh1* and

*Cox4i1* were also greater in *Nrp1*-DT cells than in *Nrp1* + DT cells (Fig. 4B). To study the functional consequences of site-directed mutations in Cox4i1 K29, Immt K312 and Mdh1 K248 from lysine to arginine at these residues, we introduced mutant plasmids into pTECs. Mutation of Cox4i1 K29 resulted in lower detectable levels of Kcr and less mitochondrial quantity (Fig. 4C, D), indicating the crotonylation of lysine residue at position 29 of Cox4i1 is crucial for maintaining mitochondrial function. However, we were unable to obtain satisfactory results for Immt and Mdh1 through Western blotting analysis due to their protein molecular weights not matching with most proteins undergoing Kcr. Consequently, our findings suggest that Nrp1, as a co-receptor of Tnr1a, promotes *Nfkb1* expression upon TNF-α stimulation, leading to the inhibition of *Etv6* transcription. This event subsequently decreases the expression of *Acox3*, which encodes an enzyme involved in the generation of crotonyl-CoA regulated by Etv6. As a result, the level of Kcr of glucose metabolism-related proteins, such as Cox4i1, a component of complex IV in OXPHOS, was lower, resulting in impaired mitochondrial function.

By scRNA-seq analysis of Kyoto Encyclopedia of Genes and Genomes (KEGG) pathways revealed significant enrichment of the apoptosis pathway in *Nrp1* + DT cells compared to *Nrp1*-DT cells. Through renal transcriptomics and pTECs proteomics, we observed lower gene and protein expression levels related to apoptosis after *Nrp1* knockout compared to the vehicle-treated controls (Fig. 4E, F). The CCK8 assay demonstrated an increased pTEC survival rate following *Nrp1* knockout compared to the vehicle group (Fig. 4G). In addition, TUNEL staining revealed greater apoptosis in pTECs after Cox4i1-K29R mutation (Fig. 4H). In addition to apoptosis, we performed scRNA-seq analysis of *Nrp1* + DT cells to investigate autophagy, pyroptosis, ferroptosis and necroptosis. We found that Nrp1 also promoted all four of these pathways in DT cells (Fig. 4I). Furthermore, by bulk RNA-seq analysis we confirmed that *Nrp1* knockout resulted in lower expression levels of genes associated with these cell death pathways (Fig. 4J), which was further verified in ultrastructure of mitochondria by electron microscope (Fig. 4K). In addition, we found that in *Nrp1* + DT cells with high expression of cell death-related genes there was a high degree of secretion of pro-fibrotic factors, such as Col4a1, Pdgfb and Pdgfa (Supplementary Fig. 8A–E).

**Nrp1 promotes renal fibrosis by inducing the secretion of collagen and the communication of DT cells with myofibroblasts**
To investigate the specific mechanism by which Nrp1 promotes kidney fibrosis, we performed further cell-cell communication analysis using scRNA-seq. We found that *Nrp1* + DT cells exhibited closer communication with other cell types compared to *Nrp1*-negative DT cells (*Nrp1*-DT), with the most significant communication occurring between *Nrp1* + DT cells and myofibroblasts (Myo) (Fig. 5A). *Nrp1* + DT cells received TGF-β signals from various cells, including myofibroblasts, while also sending PDGF signals to myofibroblasts and pericytes (Peri) (Fig. 5B). The communication between *Nrp1* + DT cells and myofibroblasts promoted production of more fibrosis-promoting factors, leading to a vicious cycle that exacerbates kidney fibrosis.

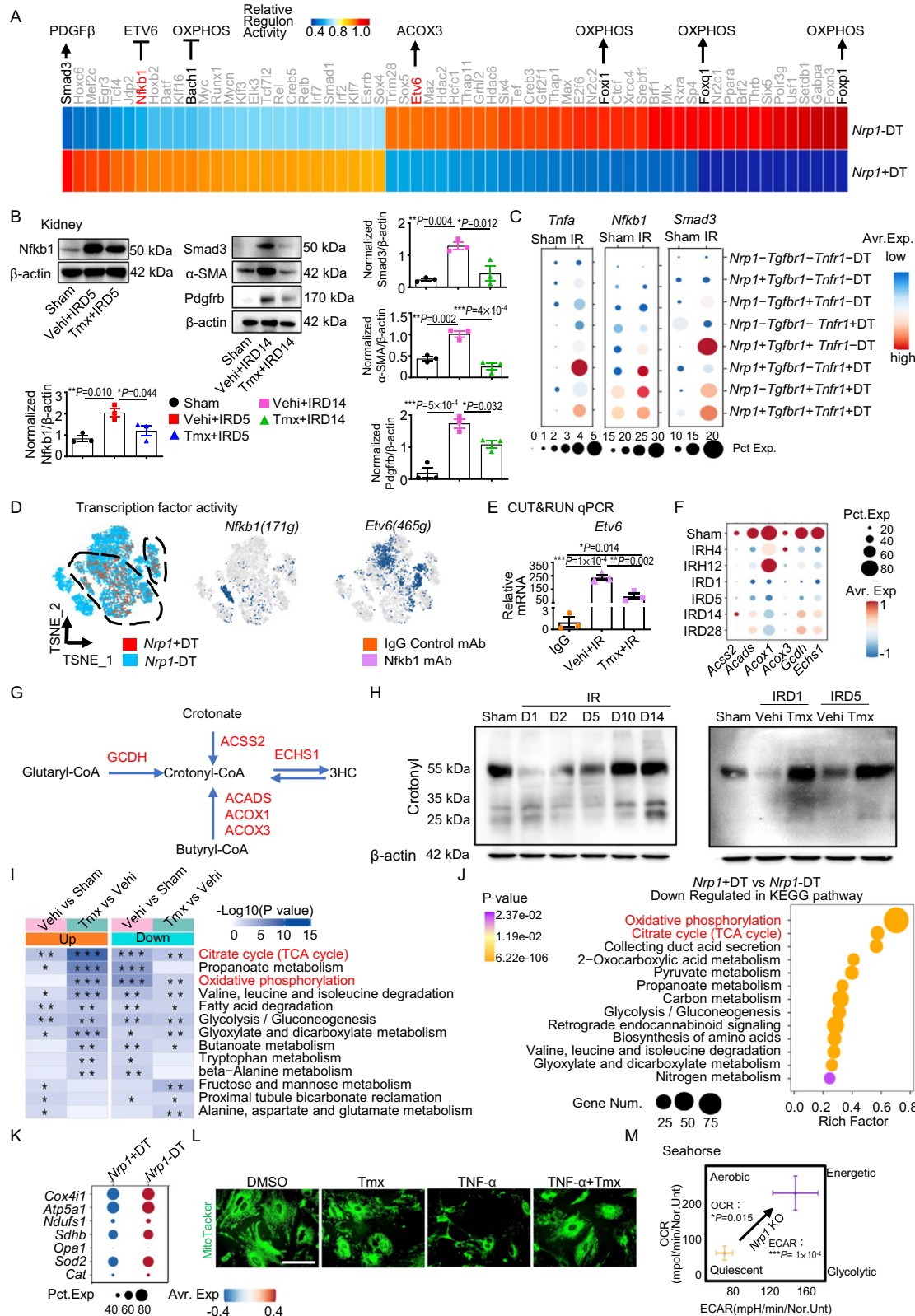

We also observed that *Nrp1* + DT cells, particularly *Nrp1*+*Tgfbr1*+ DT cells, were involved in collagen secretion. In addition to myofibroblasts, *Nrp1* + DT cells are another source of cells with high extracellular matrix (ECM) scores (Fig. 5C). By analyzing ECM-related and Myo markers-related gene sets, we found that ECM-related gene expression levels and Myo markers related gene expression levels were higher in *Nrp1* + DT cells compared to *Nrp1*-DT cells (Fig. 5D).

Furthermore, the fan-shaped bar plot revealed that *Nrp1* + DT cells exhibited greater similarity in gene expression to myofibroblasts and pericytes compared to *Nrp1*-DT cells (Fig. 5E).

Transcriptomic analysis of whole kidneys at day 5 after IR showed that the overall expression levels of ECM genes were lower when *Nrp1* was genetically deleted in renal TECs compared to non-knockout conditions (Fig. 5F, left). Similarly, proteomic data from primary renal

**Fig. 3 | Nrp1 upregulates the Nfkb1 and Smad3 pathway, suppressing *Etv6* and *Acox3* expression, leading to reduced levels of OXPHOS and TCA in TECs.** **A** Transcription factor activity analysis heatmap of DT cells. **B** Western blotting analysis showing the changes in Nfkb1, Smad3, α-SMA and Pdgfrb after IR and *Nrp1* knockout (*n* = 3 per group). **C** Dotplot of *Tnfa*, *Nfkb1* and *Smad3* in DT cells. **D** T-SNE plots showing the expression region of *Nrp1* and transcription factor activity region of Nfkb1 and Etv6 in DT cells. **E** The amplification levels of *Etv6* after binding with the anti-Nfkb1 monoclonal antibody using qPCR (*n* = 3 per group). **F** Dotplot of crotonyl-CoA-producing enzymes in kidney. **G** A schematic diagram illustrating the enzymes involved in the production of crotonyl-CoA. **H** Western blotting analysis showing the changes in crotonylation lysine modification after I-R and *Nrp1* knockout. **I** Heatmap displaying the Kyoto Encyclopedia of Genes and Genomes (KEEG) pathways enriched in proteins with changing crotonylation modification sites before and after IR treatment, as well as before and after *Nrp1* knockout. **J** Downregulated KEGG pathways in *Nrp1*+ DT cells compared to *Nrp1*-DT cells in scRNA-seq data. The statistical analyzes were two-sided and adjustments were made in *P* value. **K** Dotplot of genes related to mitochondrial functional status in DT cells. **L** Immunofluorescent staining of MitoTracker in primary renal tubular epithelial cells (pTECs) treated with TNF-α and Tmx. **M** Energy map showing increased oxygen consumption rate (OCR) and extracellular acidification rate (ECAR) in pTECs after *Nrp1* knockout (*n* = 5 per group). Data are representative of three independent experiments. \*P< 0.05, \*\*P< 0.01, \*\*\*P< 0.001 as determined by one-way ANOVA. Scale bar, 50 μm. Data represent mean ± SEM. Source data are provided as a Source Data file.

TECs (pTECs) showed that the expression levels of ECM proteins in pTECs were lower after knockout of *Nrp1* during OGD/R-induced cell injury (Fig. 5F, right). Therefore, we propose that *Nrp1* + DT cells are a TEC-type similar to myofibroblasts that spontaneously secrete collagen to promote kidney fibrosis.

Next, using the SCENIC R package to analyze transcription factors in distal TECs, we found that Smad3, which is capable of regulating the transcription of *Pdgfb*, was the transcription factor with the highest activity in *Nrp1* + DT (Fig. 3A). By examining the expression levels of *Smad3* in DT cells using a dot plot, we found that its expression was particularly high in *Nrp1*+*Tgfbr1* + DT cells (Fig. 3C). Interestingly, we found that *Smad3* was even increased in *Nrp1*+*Tgfbr1*-DT after I-R-induced injury, suggesting Nrp1 can directly promote Smad3 expression in renal TECs independently of TGF-β. Given that Smad3 activation promotes collagen and fibronectin deposition[3,39], we believe the increased transcriptional activity of Smad3 is one of the main reasons for the increased ECM score in *Nrp1* + DT cells.

Taken together, we propose that the activated *Nrp1* + TECs communicate with myofibroblasts to form a vicious cycle to promote renal fibrosis. Upon receiving TGF-β signals from myofibroblasts, Nrp1 acts either as a TGF-β co-receptor or independently to enhance the expression level and transcriptional activity of Smad3, thereby not only activating the expression of collagen-related genes, but also promoting the secretion of the pro-fibrotic factor PDGFβ, which in turn acts on myofibroblasts promoting their activation and exacerbating kidney fibrosis (Fig. 5G).

### Dual deletion of *Nrp1* and *Tgfbr1* significantly improves renal injury and fibrosis

Next, we generated mice with a single knockout of *Tgfbr1* or a double knockout of *Nrp1* and *Tgfbr1* specifically in renal TECs (Tgfbr1$^{flox/+}$,Ksp-iCre and Nrp1/Tgfbr1$^{flox/+}$,Ksp-iCre) (Supplementary Fig. 9A, B). We also generated mice with specific depletion of *Nrp1* in myofibroblasts (Nrp1$^{flox/+}$,Col1a2-iCre) (Supplementary Fig. 9C) and in pericytes (Nrp1$^{flox/+}$,Pdgfrb-iCre) (Supplementary Fig. 9D). By RT-qPCR we found a decrease in the expression levels of *Nrp1* and/or *Tgfbr1* after tamoxifen injection, confirming successful gene knockout (Supplementary Fig. 9E–H). After the gene knockout, no obvious pathological changes were observed in the heart, liver, spleen, lungs and kidneys (Supplementary Fig. 9I). We then induced I-R-mediated injury in these various knockout strains and sacrificed them at day 14 for analysis (Fig. 6A). We found that *Nrp1* and *Tgfbr1* knockout in TECs not only led to lower levels of BUN and creatine compared to the group with single knockout of *Nrp1*, but also reduced interstitial collagen deposition (Fig. 6B). Furthermore, simultaneous injection of *Nrp1* and *Tgfbr1* siRNA under the renal capsule reduced kidney injury compared to a single injection of Nrp1 siRNA (Supplementary Fig. 10A, B). By scRNA-seq analysis we found that in the DT, *Nfkb1* and *Smad3* were expressed at the highest levels in Nrp1+Tgfr1−Tnr1a+DT cells and Nrp1+Tgfr1+Tnr1a−DT cells, respectively, which was confirmed in vitro, indicating that Nrp1 interacts with different receptors, thereby activating distinct downstream pathways (Fig. 6C). Meanwhile, we found

that specific knockout of *Nrp1* in renal myofibroblasts or pericytes also resulted in less kidney fibrosis (Supplementary Fig. 10C−E).

Together, these findings suggest that Nrp1 plays multiple pro-fibrotic roles in kidney diseases. Its expression in renal TECs promotes fibrosis not only by activating the canonical TGF-β-related fibrosis pathway, but also by activating a TNF-α related pathway, and Nrp1's expression in myofibroblasts and pericytes also promotes the progression of renal fibrosis.

Overall, we conclude that Nrp1 inhibits TEC aerobic metabolism by suppressing lysine crotonylation-related modification of glucose metabolic enzymes, thus negatively affecting mitochondrial function and ultimately leading to various forms of cell death in these TECs and subsequently fibrosis (Fig. 6D)

### Discussion

Here, we investigated the impact of NRP1 on the progression of renal injury and fibrosis. We observed upregulation of NRP1 in distal TECs from kidney transplant recipients with renal dysfunction and mice with I-R-induced kidney injury. These findings suggest that NRP1 plays a crucial role in mediating renal cells death and progressive fibrosis. The expression of TNF-α is known to increase during renal injury[40], and our research revealed that TNF-α interacts with Nrp1 and Tnr1a, leading to downstream events that inhibit the crotonylation of the aerobic metabolism-related enzyme Cox4i1 through Nfkb1 activation. Furthermore, *Nrp1*-expressing DT cells secrete collagen and fibrotic factors, which activates myofibroblasts. The myofibroblasts, in turn, secrete TGF-β, which acts on TECs, inducing the secretion of more fibrotic factors. This vicious cycle aggravates the renal fibrosis. Based on in vivo and in vitro studies, we conclude that Nrp1 exacerbates kidney injury by inhibiting aerobic metabolism and promotes CKD progression by regulating myofibroblast activation and collagen secretion. Meanwhile, we explored the effects of knocking out NRP1 in tubules one day after IR, and found that the results were consistent with those observed 5 days after IR. Therefore, we believe that targeting the transmembrane cytokine co-receptor neuropilin-1 in distal tubules improves renal injury and fibrosis.

Studies have shown that IR-induced injury reduces mitochondrial quality and inhibits ATP production[41,42]. As one of the most mitochondria-rich cells in the kidney, the epithelia of the DCT is particularly susceptible to IR-mediated injury[43]. Transmission electron microscopy has revealed a decrease in mitochondria and an increase in lysosomes in DT epithelial cells after I-R, indicating ongoing cell death[44]. Increasing evidence suggests a close association between NRP1 and cell death in various disease states. For instance, NRP1 silencing has been shown to inhibit autophagy[14]. Knockout of *NRP1* in pancreatic β-cells reduces apoptosis[45]. Endothelial cell-specific loss of *Nrp1* inhibits apoptosis in mouse bone marrow endothelial cells[46]. In this study, we confirmed that Nrp1 exacerbates renal injury by inducing renal TEC death. Specifically, as a co-receptor of Tnr1a, Nrp1 binds to the receptor, activating Nfkb1 and inhibiting the crotonylation of Cox4i1 at K29, leading to mitochondrial damage and the subsequent cell apoptosis.

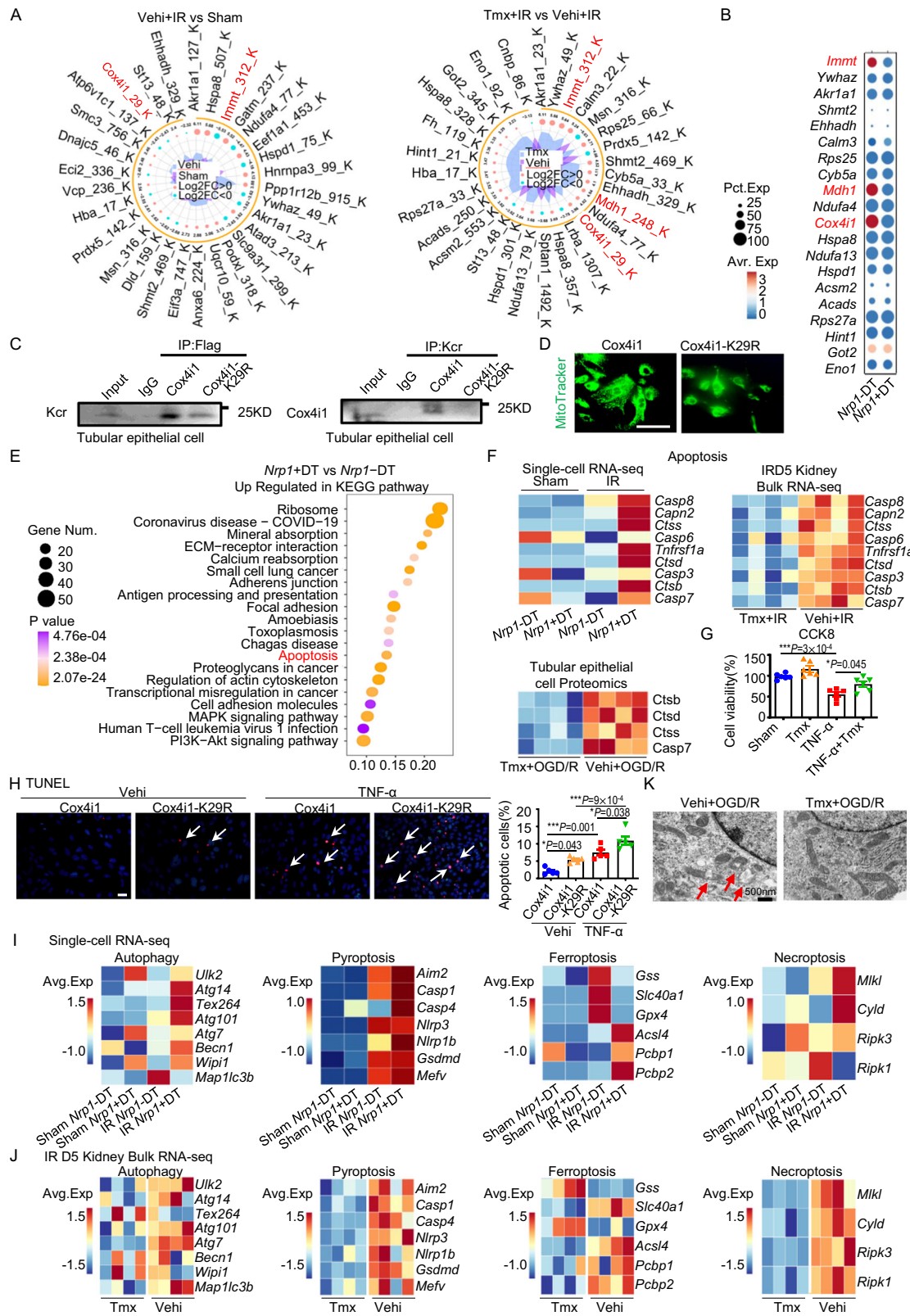

Many studies have shown a mutually reinforcing relationship between kidney cell damage and inflammation. Necrotizing cells during kidney damage release pro-inflammatory factors, triggering inflammation, which in turn promotes kidney cell death and leads to the progression of kidney disease[47,48]. However, research on the direct promotion of renal fibrosis by cell death is limited. Here, we found that *Nrp1* + DT cells express cell death-related genes and

pro-fibrotic factors, indicating that *Nrp1* + DT cell death promotes renal fibrosis.

Lysine crotonylation (Kcr) is a post-translational modification of lysine residues that has a distinct conjugated double-bond structure compared to other protein modifications[49–51]; and therefore, it has specific functions. Crotonyl-CoA, a cofactor produced during FAO and lysine/tryptophan metabolism, is directly involved in the process of

**Fig. 4 | Nrp1 reduces OXPHOS and TCA levels, exacerbating the cell death in TECs by decreasing the level of crotonylated Cox4i1. A** Radiation plot depicting the changes in crotonylation modification sites before and after IR, as well as before and after *Nrp1* knockout. **B** Dotplot showing the gene expression levels of the proteins mentioned in Figure A in DT cells. **C** Kcr level of Cox4i1 in pTECs verified by IP and Western blotting. The experiments were independently repeated three times. **D** Immunofluorescent staining of MitoTracker treated with Cox4i1 and Cox4i1-K29R plasmids. The experiments were independently repeated three times. **E** Upregulated KEGG pathways in *Nrp1* + DT cells compared to *Nrp1*- DT cells in scRNA-seq data. The statistical analyzes were two-sided and adjustments were made in *P* value. **F** Heatmaps related to apoptosis from scRNA-seq, IRD5 kidney bulk RNA sequencing, and pTECs proteomics. **G** Quantification of cell viability was detected using CCK8 (*n* = 6 per group). **H** Quantification of apoptosis were detected using terminal deoxynucleotidyl transferase dUTP nick end fluorescent labeling (TUNEL) staining (*n* = 5 per group). **I** The heatmap related to autophagy, pyroptosis, ferroptosis, and necroptosis in scRNA-seq. **J** The heatmap related to autophagy, pyroptosis, ferroptosis, and necroptosis in IRD5 kidney bulk RNA-seq. **K** The transmission electron microscopy image showed the impact of oxygen glucose deprivation/re-oxygenation (OGD/R) and *Nrp1* knockout on mitochondrial damage. The red arrow showed the damaged mitochondrial structure. The experiments were independently repeated three times. *$P$ < 0.05, **$P$ < 0.01, ***$P$ < 0.001 as determined by one-way ANOVA. Scale bar, 50 μm. Data represent mean ± SEM. Source data are provided as a Source Data file.

Kcr[38]. Enzymes involved in fatty acid and amino acid metabolism that participate in crotonyl-CoA generation promote an increase in Kcr levels[38]. Additionally, Kcr is regulated by crotonyltransferases and decrotonylases[52]. Certain enzymes, like P300/CBP, MOF and PCAF, catalyze non-histone crotonylation in mammalian cells, while class I, III and VI histone deacetylases (HDACs) serve as major decrotonylases[53]. Moreover, chromodomain Y-like protein (CDYL), which contains a crotonyl-CoA hydratase catalytic domain, hydrolyzes crotonyl-CoA and reduces Kcr[54]. Our study suggests that increased Nfkb1 transcription factor activity after I-R Inhibits *Etv6* expression, leading to decreased expression of Acox3. This further results in a reduction in renal crotonylation levels after I-R, particularly affecting lysine crotonylation sites on enzymes related to OXPHOS and the TCA. Among these enzymes, Cox4i1 is a component of cytochrome c oxidase, the final enzyme in the mitochondrial electron transport chain that drives OXPHOS. Functional studies targeting the selective regulation of Cox4i1 lysine crotonylation at K29 demonstrated that the loss of Kcr impairs Cox4i1 expression, leading to cellular dysfunction and promoting renal injury and apoptosis. Furthermore, our SCENIC analysis of transcription factor activity revealed that the *Nrp1*-DT-enriched transcription factors Foxq1 and Foxi1 promote the expression of OXPHOS-related genes, while Bach1, which is enriched in *Nrp1* + DT cells, inhibits their expression[55–57]. Additionally, we found that Foxp1, another member of the FOX family, also promotes the expression of OXPHOS components. Although as the blood supply to the medulla is scarce, the ATP in distal tubules is mainly generated by anaerobic glycolysis[58], recent studies have found that OXPHOS disruption also contributes to distal tubular damage[59,60]. Thus, our finding suggests NRP1 related-aerobic metabolism disturbance is one of the mechanisms of NRP1 induced distal tubular injury.

Nrp1 is closely associated with fibrosis and has been implicated in promoting fibrosis in various organs, including the liver and lung, as well as in the context of pancreatic ductal adenocarcinoma[16,18,20,61]. In our study, we found that *Nrp1* + DT cells directly secrete collagen, contributing to fibrosis. Moreover, upon receiving TGF-β signals, Nrp1 cooperates with Tgfr1 to activate Smad3 signaling in distal renal TECs, leading to the secretion of Pdgf, which, in turn, further activates myofibroblasts. Simultaneously, activated myofibroblasts secrete more TGF-β, forming a detrimental cycle that drives continuous renal fibrosis progression. Through the construction of genetically modified mice and in vitro experiments, we demonstrated that deletion of Nrp1 or simultaneous deletion of the genes encoding the TGF-β receptor and Nrp1 delay the progression of renal fibrosis and inhibit Smad3 expression. Notably, the combined deletion of both receptors yielded a more pronounced effect. Meanwhile, specific knockout of *Nrp1* in renal myofibroblasts or pericytes also reduced kidney fibrosis. In addition, we verified that the expression of Nfkb1 increases when *Nrp1* and TNF-α receptors are simultaneously expressed in distal renal TECs. Similarly, the expression of Smad3 increases when Nrp1 and TGF-β receptors are co-expressed. Nrp1 interacts with different receptors to promote the activation of distinct signaling pathways, thereby exacerbating renal injury and fibrosis. These findings suggest that Nrp1 plays a pro-fibrotic role in various cell types after renal injury.

In this study, we comprehensively investigated the role of Nrp1 in both AKI and CKD. Firstly, we confirmed that Nrp1 is expressed in DT cells, and its expression is upregulated following I-R. Knockout of Nrp1 in DT improved IR-induced renal injury and subsequent renal fibrosis. Secondly, we demonstrated that Nrp1 interacts with the TNF-α receptor Tnr1a, influencing downstream signaling pathways, ultimately leading to decreased Kcr of Cox4i1 in renal tubular epithelial cells. This inhibition of aerobic metabolism in renal tubular epithelial cells led to cell death. Finally, we innovatively discovered that *Nrp1* + DT possess pro-fibrotic properties by secreting collagen. Meanwhile, we further confirmed that the communication between *Nrp1* + DT and myofibroblasts establishes a vicious cycle of mutual activation through reciprocal secretion of pro-fibrotic factors, accelerating renal fibrosis.

In conclusion, our study highlights the significant involvement of Nrp1 in the advancement of renal diseases (Fig. 6D). Nrp1 interacts with both TGF-β and TNF-α receptors in distal TECs while activating the Nfkb1 and Smad3 pathway, regulating the secretion of collagen, suppressing OXPHOS, activating myofibroblasts, thereby accelerating the progression of acute and chronic kidney diseases. Blockade of Nrp1 and the TGF-β receptor in distal tubules significantly improved renal injury and renal fibrosis, which present a therapeutic target for addressing AKI and its progression to CKD.

## Methods

### Ethics statement

Hospitalization information from patients with renal transplant dysfunction was collected from Tongji Hospital, Tongji Medical College, Huazhong University of Science and Technology. A total of 217 patients were included in the study, and the main clinical characteristics collected were gender, age, history of hypertension, history of diabetes, plasma creatinine, blood urea nitrogen and eGFR levels. Based on the staging of chronic kidney disease using eGFR, the data was divided into two groups: the severe decline group (115 cases) with an eGFR <30 ml/min/1.73 m², and the mild-to-moderate decline group (102 cases) with an eGFR ≥ 30 ml/min/1.73 m². Corresponding paraffin-embedded tissue sections of human kidneys were collected from renal biopsy samples of patients with renal transplant dysfunction. Human tissue samples were obtained from individuals who provided informed consent. The study was conducted in accordance with the principles of the Helsinki Declaration and obtained approval from the Medical Ethics Committee of Tongji Hospital, Tongji Medical College, Huazhong University of Science and Technology (TJ-IRB20230446), and informed consent was obtained from the patients.

Animal care and experimental procedures were approved by the Animal Ethics Committee of Huazhong University of Science and Technology ([2023] IACUC number: 3181).

### Mice

Male pathogen-free C57BL/6 mice (7-8 weeks old, weighing 18-22 grams) were purchased from Beijing Huafu Kang Biotechnology Co.,

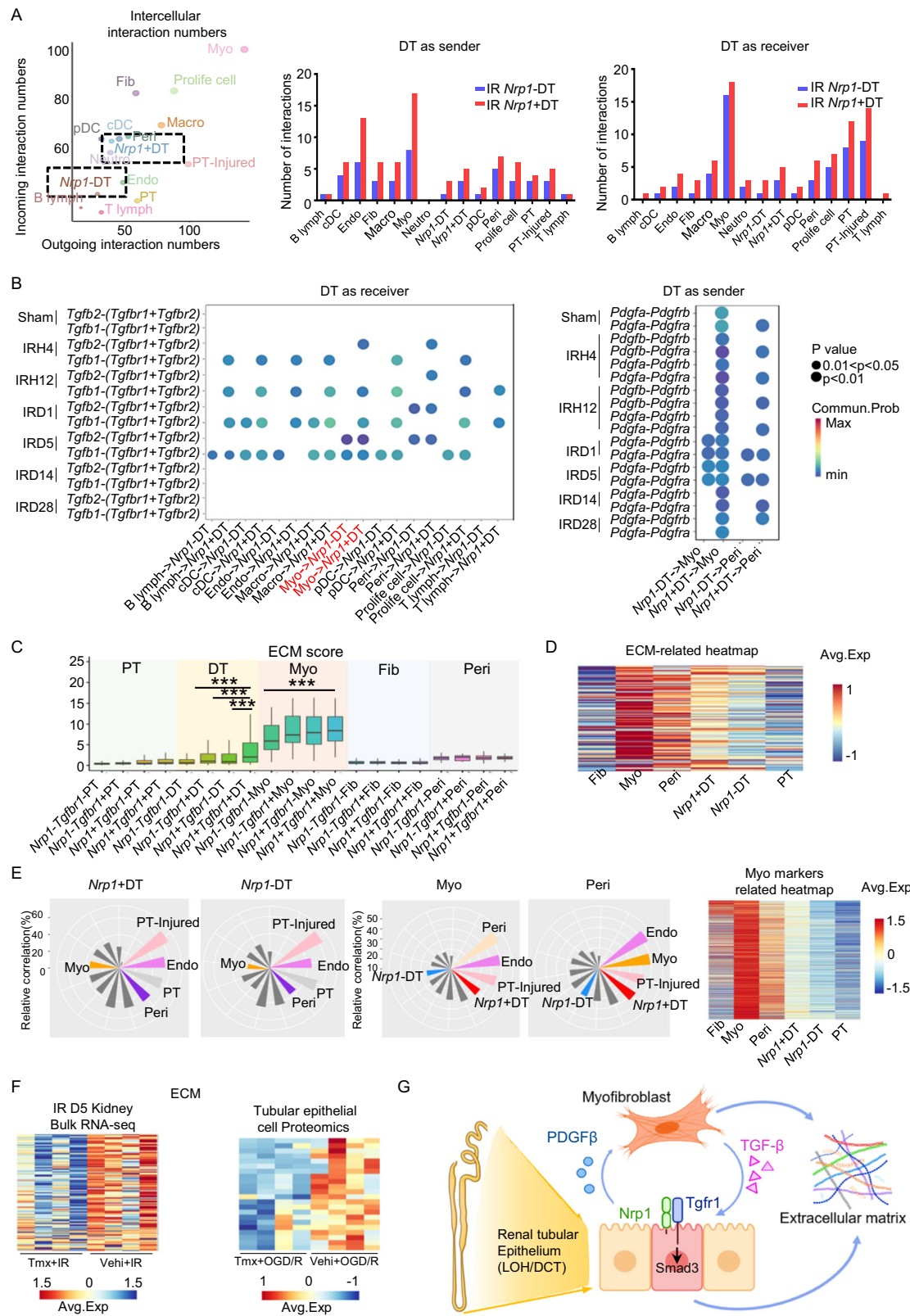

Ltd. Nrp1-loxp mice were obtained from Shanghai Model Organisms Center, Inc. Tgfbr1-loxp, Col1a2-iCreERT and Pdgfrb-iCreERT mice were purchased from Cyagen Biosciences in China. KSP-iCreERT mice were developed in collaboration with BioCytogen using CRISPR/Cas9 technology. All mice were housed in a specific pathogen-free and sterile environment at the Animal Experimental Center of Tongji Medical College, Huazhong University of Science and Technology. All

mice were fed with normal chow diet (about 4.2% fat and 21% protein, WQJX Bio-Technology). The mice were maintained at 22 °C temperature and 50% humidity conditions, following a 12-hour light-dark cycle. C57BL/6 mice underwent a one-week acclimation period prior to the experiments. The housing and experimental procedures for the mice strictly adhered to the guidelines of the National Institutes of Health (NIH) and the Animal Management Committee of Tongji Medical

**Fig. 5 | *Nrp1* + DT cells communicate with myofibroblasts and secrete collagen to promote renal fibrosis. A** Scatter plot and bar chart of renal intercellular interaction numbers after I-R surgery. **B** Dotplot showing DT cells receiving TGF-β signals and sending PDGF signals. The statistical analyzes were two-sided and adjustments were made in *P* value. **C** The extracellular matrix (ECM) scores of renal tubular epithelial cells and stromal cells (*n* = 10 per group). In the boxplots, the central line represents median, the bounds of boxes represent the first and third quartiles, and the upper and lower whiskers extend to the highest or the smallest value within 1.5 interquartile range. **D** The ECM-related and myofibroblasts markers-related heatmap of renal tubular epithelial cells and stromal cells. **E** Gene expression similarity between *Nrp1* + DT and other cell types was presented by fan-shaped bar plot. **F** ECM-related heatmap in IRD5 Kidney Bulk RNA-seq and pTECs Proteomics (*n* = 4 per group). **G** A schematic diagram illustrating the role of Nrp1 in promoting fibrosis. Figure 5G, created with BioRender.com, released under a Creative Commons Attribution-NonCommercial-NoDerivs 4.0 International license. *$P < 0.05$, **$P < 0.01$, ***$P < 0.001$ as determined by one-way ANOVA. Data represent mean ± SEM. Source data are provided as a Source Data file.

College, Huazhong University of Science and Technology. All mice used in this study were males, aged 8-12 weeks. All mice were assigned to groups randomly.

### Animal model

The procedure for the bilateral renal ischemia-reperfusion (I-R) injury model is as follows: under sterile conditions, the body temperature of the mice was regulated to 36.6-37.2 °C using a temperature controller (from FHC, USA). The mice renal hilum was clamped using mouse arterial clamps (from Roboz Surgical Instrument Co, Germany) for 30 min. The color of the kidneys was observed before and after clamping to ensure the success of the ischemia-reperfusion procedure. After renal blood flow was restored, the incision was sutured.

The unilateral ureteral obstruction (UUO) model: The left ureter was exposed and isolated, ligated at the level of the lower pole of the left kidney using 4-0 silk suture, and then transected. The wound was closed layer by layer.

5/6 nephrectomy (5/6 Nx) model: The adrenal gland was separated from the kidney, followed by dissecting the renal pedicle and perirenal fat tissue. Then, the upper and lower poles of the left kidney were removed, and one week later, the right kidney was also removed.

### Renal function

The plasma creatinine concentration was measured using the QuantiChrom™ Creatinine Assay Kit (BioAssay Systems, USA), following the manufacturer's instructions.

The plasma blood urea nitrogen (BUN) concentration was measured using the QuantiChrom™ Urea Assay Kit (BioAssay Systems, USA), following the manufacturer's instructions.

### Histology and immunofluorescence

The kidneys were fixed in 4% paraformaldehyde for 24 h and subsequently embedded in paraffin. Renal pathology was evaluated by performing Periodic Acid-Schiff (PAS) staining. The degree of renal fibrosis was assessed using Masson and Sirius Red staining. Heart, liver, spleen, and lung tissues were stained with hematoxylin and eosin (H&E) to evaluate pathological changes. Ten randomly selected fields were quantitatively evaluated by two blinded renal pathologists. Data analysis was performed using Image J software from the National Institutes of Health (NIH), USA.

For immunofluorescence (IF), kidney sections were deparaffinized at 70 °C and underwent antigen retrieval at 98 °C for 30 min using a microwave oven. Non-specific antigen blocking was performed with serum at room temperature for 30 min. The sections were then incubated overnight at 4 °C with specific antibodies, including Nrp1 (1:100, Abcam, UK, ab81321; R&D Systems, USA, FAB5994P; Santa Cruz Biotechnology, USA, sc-5307), Kim1 (1:1000, R&D Systems, USA, AF1817), S12a3 (1:100, Abcam, UK, ab95302), Tgfr1 (1:100, Proteintech, China, 30117-1-AP), and Tnr1a (1:100, Abcam, UK, ab223352). Immunofluorescence secondary antibodies were used for labeling. Cell nuclei were stained with DAPI. Ten fields were randomly selected and quantified on each slide by two experienced renal pathologists using blind assessment, and data analysis was performed using Image J software (NIH, USA).

### Fluorescence in situ hybridization (FISH) and immunofluorescence dual labeling staining

The *Nrp1* probe was provided by Wuhan Servicebio Technology Co., Ltd (China). The specific probe sequence was detailed in Table S2. The steps for Fluorescence in Situ Hybridization (FISH) and Immunofluorescence dual labeling staining were as follows: After deparaffinizing and hydrating the paraffin sections, the sections were boiled in fixing solution for 15 min. Proteinase K (20 μg/ml) was added for 20-30 min. After rinsing with water, PBS washing was performed three times for 5 min each. Prehybridization solution was added and incubated at 37 °C for 1 h. The prehybridization solution was removed, and the hybridization solution containing the probe was added, followed by overnight hybridization at 37 °C. After washing the hybridization solution away, the primary antibody Cdh16 (1:100, Proteintech, China, 15107-1-AP) was added and incubated overnight at 4 °C. The secondary antibody was added and incubated at room temperature for 60 min. The sections were then incubated with DAPI in the dark for 8 min. The experimental process was conducted under conditions devoid of nucleases.

### Western blotting

Proteins were extracted from renal tissue, and equal amounts of protein (30 μg) from different samples were separated by SDS-PAGE and transferred onto PVDF membranes. The membranes were blocked in blocking buffer (5% BSA in TBST) at room temperature for 1 h, and then incubated overnight at 4 °C with primary antibodies against Nrp1 (1:1000, Abcam, UK, ab81321), Tnr1a (1:1000, Abcam, UK, ab223352), Cox4i1 (1:1000, CST, 4850), Nfkb1 (1:1000, CST, 13586), Smad3 (CST, 1:1000, 9523), Pdgfrb (1:1000, Abcam, UK, ab69506), α-SMA (1:1000, Abcam, UK, ab7817) and β-actin (1:1000, CST, 4967), as well as the pan-antibodies against crotonylation (PTM-502), 2-hydroxy-isobutyrylation (PTM-802), succinylation (PTM-419), acetylation (PTM-105RM), β-hydroxy-butyrylation (PTM-1201RM), lactylation (PTM-1401RM), and malonylation (PTM-902) (1:1000) provided by Jingjie PTM BioLab. The PVDF membranes were then incubated with HRP-conjugated secondary antibodies at 37 °C for 1 h, followed by visualization using an enhanced chemiluminescence method (ECL, Biosharp, China). The signal intensity of the target bands was quantitatively analyzed using Image J software (NIH, USA).

### Enzyme linked immunosorbent assay (ELISA)

The plasma Kim1 concentration was measured using the Mouse KIM1 ELISA Kit (Boster Biological Technology, China), following the manufacturer's instructions. The specific steps are as follows: add samples and standards, react at 37 °C for 90 min, add biotinylated antibody, and react at 37 °C for 60 min. Wash with wash buffer solution three times. Add ABC and react at 37 °C for 30 min. Wash with wash buffer solution 5 times. React at TMB 37 °C for 15 min. Finally, add the stop solution and measure the OD value at 450 nm.

### Quantitative real time-PCR

RNA from renal tissue or cells was extracted using Trizol reagent (Invitrogen, USA) following the manufacturer's instructions. The extracted RNA was then reverse transcribed into single-stranded cDNA using a reverse transcription system (Vazyme, China). Quantitative

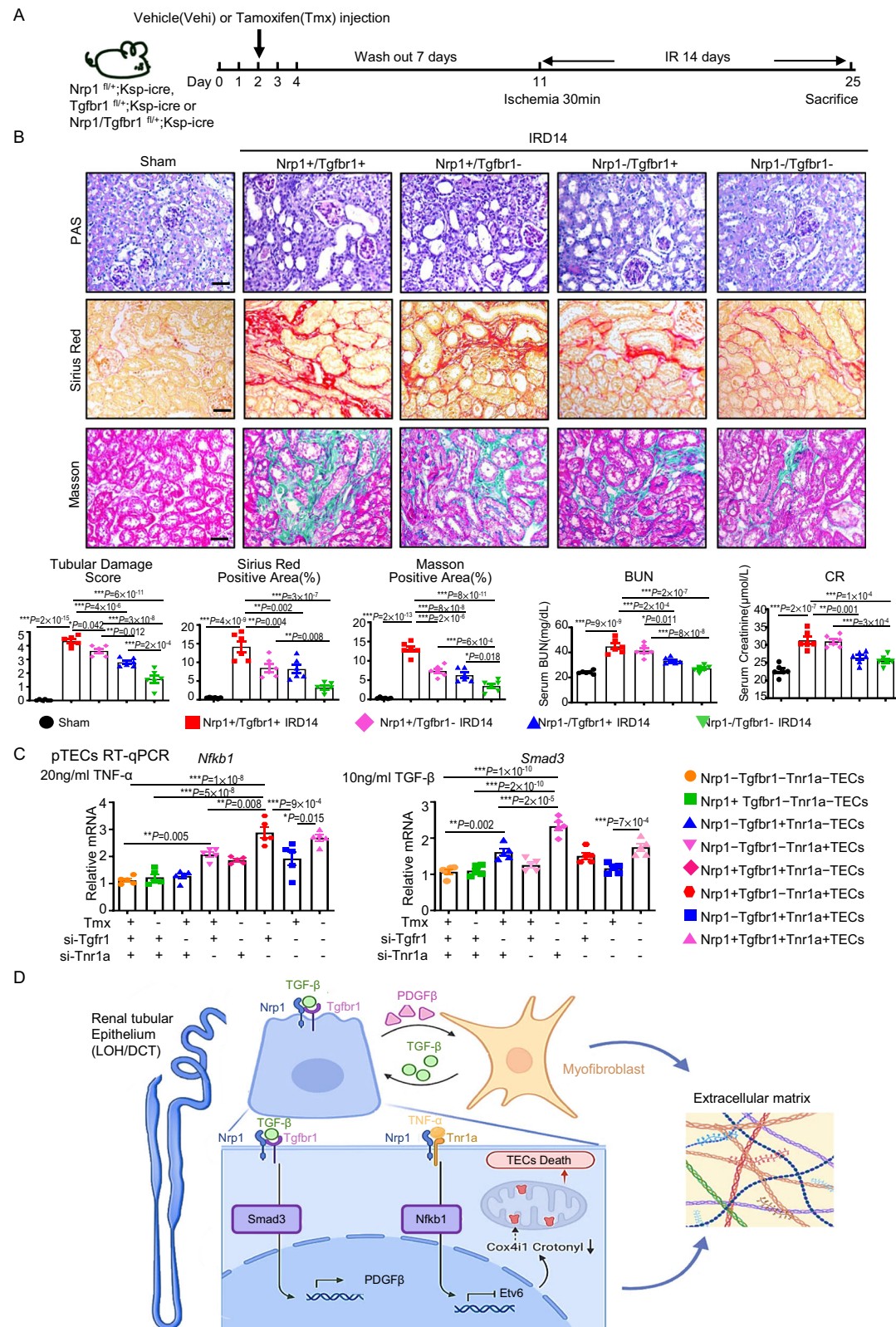

**Fig. 6 | Dual knockout of *Nrp1* and *Tgfbr1* in distal TECs better improves renal injury and renal fibrosis compared to either knockout alone. A** A schematic diagram illustrating the experimental scheme. **B** Representative micrographs and corresponding statistical scores of PAS, Masson, and Sirius red staining on day 14 after I-R in mice with tubular-specific *Nrp1* and *Tgfbr1* knockout. Plasma BUN concentrations and CR concentrations in sham, vehicle + IR, or Tmx + IR groups at 14 days (*n* = 6 per group). **C** Expression levels of *Nfkb1* and *Smad3* in pTECs treated with TNF-α or TGF-β by RT-qPCR (*n* = 5 per group). **D** A schematic diagram illustrating the mechanism by which Nrp1 promotes kidney injury and fibrosis. Figure 6D, created with BioRender.com, released under a Creative Commons Attribution-NonCommercial-NoDerivs 4.0 International license. *\*P* < 0.05, *\*\*P* < 0.01, *\*\*\*P* < 0.001 as determined by one-way ANOVA. Scale bar, 20 μm. Data represent mean ± SEM. Source data are provided as a Source Data file.

PCR was performed using SYBR master mix (Vazyme, China) on a Step-One real-time PCR system (ABI, USA). The relative mRNA expression levels were calculated using the $2^{-\Delta\Delta Ct}$ method and normalized to the expression level of *Gapdh*. The primer sequences used are listed in Table S3.

## Immunoprecipitation

The target cells or tissues were lysed using NP-40 buffer, with the addition of protease inhibitors and phosphatase inhibitors to prevent protein degradation and phosphorylation. After centrifugation, the supernatant was transferred to a new centrifuge tube. Specific antibodies were added to the supernatant to bind to the target proteins, forming antigen-antibody complexes. The duration and temperature of immunoprecipitation were optimized based on experimental requirements. The antigen-antibody complexes were then bound to Protein A/G Magnetic Beads, and the beads were washed multiple times with NP-40 buffer to remove non-specific proteins and impurities. The target proteins were eluted from the beads using SDS sample buffer. The eluted samples could be further analyzed using Western blotting experiments.

## Isolation and culture of primary renal tubular epithelial cells (pTECs)

After euthanizing the mice, the kidneys were isolated in a sterile laminar flow hood. The kidneys were then cut into approximately $1\,mm^3$ pieces and placed in 0.2% collagenase IV at 37 °C for 1 h in a cell culture incubator. After digestion, the tissue was filtered through sterile sieves with pore sizes of 70 μm and 40 μm to obtain a single-cell suspension. The suspension was centrifuged to pellet the cells, and the red blood cells were lysed. The cell pellet was resuspended and transferred to Falcon Bacteriological Petri Dishes (351029, Corning), and incubated in cell culture incubator for 1 h to allow adherent cells to settle at the bottom of the culture dish. Afterwards, the supernatant containing non-adherent cells was transferred to Falcon 100 mm TC-treated Cell Culture Dishes (353003, Corning), and placed in cell culture incubator for further culturing.

## Oxygen glucose deprivation/reoxygenation

To culture primary renal tubular epithelial cells, DMEM/F12 medium was used to maintain an appropriate cell number and growth state. After overnight starvation in medium without growth supplements, the medium was replaced with glucose-free medium (Gibco, 11966-025). The cultured cells were then placed in a hypoxia incubator with 1% $O_2$ for 6 h. Following the hypoxia treatment, the cells were returned to normal oxygen and regular culture medium conditions for 6 h. Cell samples were collected for subsequent experiments and analysis.

## Metabolic Measurement

To measure the real-time oxygen consumption rate (OCR) and extracellular acidification rate (ECAR), the Seahorse XFe24 Analyzer (Agilent) was utilized. Primary renal tubular epithelial cells obtained from Nrp1 $^{flox/+}$;Ksp-icre mice were isolated and plated in each well of the XF24 cell culture microplate. The cells were then starved overnight in DMEM/F12 medium without growth supplements. Subsequently, the cells were treated with 4-OHT to knockout Nrp1 or its solvent DMSO, in a complete growth medium for 24 h, followed by oxygen-glucose deprivation/reoxygenation conditions.

One hour prior to the real-time measurement, the medium was switched to Seahorse XF DMEM assay medium (103680-100, Agilent), and specific drugs such as oligomycin, FCCP, and rotenone/antimycin A were added according to the provided instructions. Once the Seahorse measurement was completed, the cell numbers in each well were counted, and the OCR and ECAR readings were normalized using the estimated average cell number. Four distinct cellular energy states

(quiescent, aerobic, glycolytic, and Energetic) were defined based on existing literature[62] and the manufacturer's instructions.

## Transcriptome sequencing and bioinformatics analysis

The total RNA was extracted from kidney tissue using Trizol reagent (Invitrogen, USA) by disrupting and purifying the tissue. The mRNA was reverse transcribed into cDNA and subsequently amplified. Subsequently, the PCR products were converted into single-stranded circular products, which then formed DNA nanoballs (DNBs) containing multiple copies of the DNA. The sequencing was performed using the combinatorial Probe-Anchor Synthesis (cPAS) technology. The raw data obtained from sequencing were subjected to quality assessment and filtering using SOAPnuke (v1.5.6) to obtain clean data. The clean data was mapped to the mouse mm9 genome using HISAT (Hierarchical Indexing for Spliced Alignment of Transcripts). The clean data was aligned to the reference gene set using Bowtie2 (v2.3.4.3). Gene expression quantification was performed using the RSEM (v1.3.1) software. Differential gene expression analysis was conducted using DESeq2 (v1.4.5). Downstream bioinformatics analysis was performed using R software packages to generate results and gain insights.

## TMT quantification proteomics and bioinformatics analysis

Four samples per group (a total of eight samples) were used for analysis. After extracting proteins and quantifying them, the proteins were enzymatically digested into peptides and labeled with TMT reagents. Peptide separation was performed using a Shimadzu LC-20AD liquid chromatography system. Following liquid phase separation, the peptides were subjected to mass spectrometry analysis in DDA (Data Dependent Acquisition) mode using the Orbitrap Exploris 480 tandem mass spectrometer (Thermo Fisher Scientific, San Jose, CA). The raw mass spectrometry data was then converted into mgf format using appropriate tools and searched against the uniprot_mmu databases using protein identification software such as Mascot. Quality control analysis was conducted simultaneously to determine the suitability of the data. Subsequently, TMT-based quantitative analysis was performed to identify significantly differentially expressed proteins of interest. Finally, downstream bioinformatics analysis was conducted.

## Proteomic experiment of protein modifications

The modified proteomics experiment was conducted by Jingjie PTM BioLab Ltd.(Hangzhou, China). In the experiment, three mice were used in each group, and the kidney from the three mice were mixed into one sample for analysis. After collecting kidney tissues, proteins were extracted and enzymatically digested with trypsin. The resulting peptides were dissolved in IP buffer (100 mM NaCl, 1 mM EDTA, 50 mM Tris-HCl, 0.5% NP-40, pH 8.0) and transferred to pre-washed resin for overnight incubation at 4 °C. After incubation, the resin-bound peptides were eluted using 0.1% trifluoroacetic acid and then vacuum dried to enrich the modified peptides. The peptides were dissolved in the mobile phase of liquid chromatography and separated using the NanoElute ultra-high-performance liquid chromatography system. The separated peptides were ionized in the Capillary ion source and analyzed using the timsTOF Pro 2 mass spectrometer (Bruker) to achieve liquid chromatography-tandem mass spectrometry analysis. The Raw files obtained from mass spectrometry detection were subjected to bioinformatics analysis using software.

Modified proteomics detected the changes in lysine crotonylation modification sites of proteins between different groups. However, these changes could be due to the increase or decrease of the proteins themselves. By simultaneously performing proteomics and modified proteomics analysis on the samples, the relative changes in lysine crotonylation modification sites of proteins could be determined.

## Terminal deoxynucleotidyl transferase dUTP nick end fluorescent labeling assay

The primary renal tubular epithelial cells were subjected to terminal deoxynucleotidyl transferase dUTP nick-end labeling (TUNEL) staining using the CoraLite 594 TUNEL Cell Apoptosis Detection Kit (Proteintech, China), following the manufacturer's instructions. The stained cell slides were observed under a fluorescence microscope at the appropriate wavelength. Typical images that best represent the average values in the quantitative analysis were selected.

## Live-cell mitochondrial fluorescence staining

The primary renal tubular epithelial cells were subjected to live-cell mitochondrial fluorescence staining using MitoTracker Green (Beyotime, China), following the manufacturer's instructions. The stained cells were captured using a fluorescence microscope at the appropriate wavelength. Typical images that best represent the average values in the quantitative analysis were selected.

## Hyperactive pG-MNase CUT&RUN assay kit for qPCR

To investigate protein-DNA interactions in primary renal tubular epithelial cells, the CUT&RUN (Cleavage Under Targets and Release Using Nuclease) Assay Kit (HD101, Vazyme, China) was used. The CUT&RUN technique utilizes MNase (Micrococcal Nuclease) fused with Protein G, which allowed precise targeting of the desired protein and cleavage of DNA fragments near the target sites under antibody guidance. In brief, Concanavalin A-coated Magnetic Beads Pro, which were conjugated with Protein A, were first used to capture the cells. The cells were then permeabilized using a non-ionic detergent, digitonin, to allow access to the cell membrane. With the aid of primary antibodies specific to the target protein and Protein G, the MNase enzyme fused with Protein G was guided to cleave the DNA sequences near the target protein. qPCR analysis of the cleaved DNA fragments provided information about the protein-DNA interactions. The experiment was conducted following the manufacturer's instructions.

## Construction of plasmids and cell transfection

The plasmids used in this study were packaged and prepared by GenScript Biotech Corporation (Shanghai, China). The mutant plasmid pcDNA3.1(+)-Cox4i1(K29R)-3flag was introduced into renal tubular epithelial cells to investigate the protein interaction status and cellular functional analysis of Cox4i1 after the K29 mutation.

## Subcapsular injection of lentivirus for NRP1 overexpression

The NRP1 overexpression lentivirus used in this study was packaged and prepared by GenScript Biotech Corporation (Shanghai, China). In vivo experiments involved the subcapsular injection of lentivirus into the kidneys of C57 mice, following the manufacturer's instructions. Ten days later, ischemia-reperfusion (IR) was induced, and after 5 days of IR, the mice were euthanized for subsequent experimental analysis.

## Tgfr1, Tnr1a and Nrp1 siRNA transfection

In vitro, primary renal tubular epithelial cells were cultured to 50-60% confluency and transfected with Tgfr1 or Tnr1a siRNA (RiboBio, China) using Lipofectamine RNAiMAX (Life Technologies) according to the manufacturer's instructions. On the third day post-transfection, cells were collected for protein and RNA extraction to validate the gene silencing effect. The mRNA samples from the third day were used for experimentation and analysis. In vivo, Tgfr1 or/and Nrp1 siRNA was injected into subcapsular space of the kidneys of C57 mice. Three days later, IR was induced, and after 5 days of IR, the mice were euthanized for subsequent experimental analysis.

## Preparation of single cell suspension

After euthanizing the mice, chilled PBS (phosphate-buffered saline) was injected into the left ventricle of the heart. The kidneys were then dissected and cut into small pieces of approximately 1 mm³. The Multi Tissue dissociation kit (Miltenyi, 130-110-203) was used for digestion, incubating at 37 °C for 30 min. The solution was then passed through a 40 μm cell strainer. After centrifugation at 1000 g for 5 min, the cell pellet was resuspended in 1 ml of red blood cell lysis buffer and incubated on ice for 3 min. The cell number and viability were analyzed using the Countess AutoCounter (Invitrogen, C10227). This method generated a single-cell suspension with over 85% viability.

## Single-cell RNA-sequencing and data preprocessing

The single-cell suspension, 10x barcoded gel beads, and oil droplets were added to the wells of the 10x Chromium Single Cell instrument (10x Genomics) according to the manufacturer's instructions, forming a droplet-in-oil structure to capture individual cells. The process involved Gel Bead-In-EMulsions (GEMs) generation, barcode processing, GEM-RT cleanup, cDNA amplification, and library construction, as described by the manufacturer. The GEMs containing cDNA were then broken, and the cDNA libraries from all individual cells were pooled together. Prior to pooling, the libraries were quantified using Qubit. Unique molecular identifiers (UMIs) were also added to identify PCR duplicates. The final mixed libraries for each experiment were sequenced on the Illumina Novaseq 6000 platform at the Wuhan Institute of Biotechnology. The sequencing results were demultiplexed and converted to FASTQ format using the Illumina bcl2fastq software. The Cell Ranger Single-Cell Software Suite was used for sample demultiplexing, barcode processing, and single-cell 3' gene counting. The cDNA inserts were aligned to the mm10/GRCm38 reference genome. Only reliably mapped non-PCR duplicate sequences with valid barcodes and UMIs were used to generate a gene-barcode matrix. The samples included seven groups: Sham, consisting of two samples (ShamA and ShamB); IR H4, consisting of one sample; IR H12, consisting of one sample; IRD1, consisting of one sample; IRD5, consisting of three samples (IRD5A, IRD5B, and IRD5C); IRD14, consisting of one sample; and IRD28, consisting of one sample.

## Single-cell data analysis

The further analysis was performed using the Seurat R package (version 4.2.0), including quality filtering, identification of highly variable genes, dimensionality reduction, standard unsupervised clustering algorithms, and discovery of differentially expressed genes. To exclude low-quality cells, we removed cells with low gene counts (<200) and cells with extreme gene expression that may represent multiple cells or doublets. We also removed cells with high transcription of mitochondrial genes (>50%). After quality control filtering, a total of 101,683 cells were retained for subsequent analysis (ShamA: 9962 cells, ShamB: 6,860 cells, IRH4: 11,794 cells, IRH12: 11,973 cells, IRD1: 9,251 cells, IRD5A: 13,592 cells, IRD5B: 8,721 cells, IRD5C: 6,797 cells, IRD14: 9,244 cells, IRD28: 13,489 cells). The top 2000 highly variable genes were determined for integration analysis based on variance/mean across all genes. Principal component analysis (PCA) was then performed using these variable genes, and significant principal components (PCs) were determined using the jackStraw function in the Seurat package. The clustering results were visualized using UMAP. Differential gene expression analysis was conducted using the FindMarkers function in Seurat. Gene set enrichment analysis was performed using the ClusterProfiler package. Cell proportions and gene expression visualization analysis were performed using the Plot1cell package. Cell-cell communication analysis was performed using Cellchat package. Transcription factor analysis was performed using SCENIC package.

## Statistical analysis

Single-cell data were analyzed using R 4.2.1, and the other statistical analyses were analyzed using GraphPad Prism software version 8.0.2 or IBM SPSS Statistics 26. Normally distributed data are presented as

mean ± SEM, while non-normally distributed data are presented as median and quartiles. Chi-square test was used for comparing ratios. For normally distributed data with one variable, unpaired Student's *t*-test (two-tailed) was used for two-group comparisons, and 1-way ANOVA followed by Tukey's multiple comparison test was used for multiple-group comparisons. For non-normally distributed data, Mann–Whitney U test (two-tailed) was used for two-group comparisons. Univariate binary logistic regression was used to select and evaluate clinically relevant variables associated with eGFR and the relationship between NRP1 and eGFR. The selected variables from the univariate analysis were included in the multivariate logistic regression model for further analysis, with the inclusion criterion of $P < 0.05$ and exclusion criterion of $P \geq 0.05$. The results were presented as odds ratios (ORs), 95% confidence intervals (CIs), and P values. All statistical analyzes were two-sided. *$P < 0.05$, **$P < 0.01$, ***$P < 0.001$.

## Reporting summary
Further information on research design is available in the Nature Portfolio Reporting Summary linked to this article.

## Data availability
The raw bulk RNA-seq data reported in this paper have been deposited in the Genome Sequence Archive[63] in National Genomics Data Center[64], China National Center for Bioinformation / Beijing Institute of Genomics, Chinese Academy of Sciences (GSA: CRA016479). The pTECs proteomics data have been deposited to the ProteomeXchange Consortium via the PRIDE[65] partner repository with the dataset identifier PXD052244. The kidney proteomics data and modified proteomics detected the changes in Kcr have been deposited to the ProteomeXchange Consortium via the PRIDE[65] partner repository with the dataset identifier PXD052293. The scRNA-seq data used in this study are available in the Genome Sequence Archive[63] in National Genomics Data Center[64], China National Center for Bioinformation / Beijing Institute of Genomics, Chinese Academy of Sciences (GSA: CRA017133) according to the recommend on https://www.springernature.com/gp/authors/research-data-policy/biological-sciences-repositories/12327160. Source data are provided with this paper. All other data needed to reproduce the results presented here can be found in the manuscript, figures and supplementary materials. Source data are provided with this paper.

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

## Acknowledgements

The authors express sincere gratitude to all individuals involved in this study. This research was financially supported by the National Natural Science Foundation of China (Grants No. 82170701 to Y.Y, 81974087 to Y.Y, 82370700 to Y.Y and 82370699 to R.Z.), the Major Research plan of the National Natural Science Foundation of China (Grant No. 82230021 to G.X.) and the National Natural Science Foundation of China for Young Scholars (Grant No. 82200768 to H.Z.). We have obtained permission from Benjamin Humphreys MD, PhD to use the Fig. 1C images obtained from http://humphreyslab.com/SingleCell/.

## Author contributions

G.X., Y.Y. and R.Z. designed the study; Y. Li. and Z.W. performed the bioinformatics analysis.; H.X., M.W., C.C. and H.Z. collected and analyzed the clinical data; Y. Li., Y.H., M.S., B.H., X.W. and S.M. performed or analyzed animal experiments; Y. Li., D.H., C.X. and Y. Lin performed or analyzed in vitro experiments; Y. Li. and R.Z. wrote and discussed the paper; G.X., Y.Y., and R.Z. conceived the project and supervised and coordinated all the work.

## Competing interests

The authors declare no competing interests.
