## [Peer Review File · Nature Communications]

Targeting the transmembrane cytokine co-receptor neuropilin-1 in distal tubules improves renal injury and fibrosisEditorial note: Parts of this Peer Review File have been redacted as indicated to maintain the confidentiality of unpublished data. Parts of this Peer Review File have been redacted as indicated to remove third-party material where no permission to publish could be obtained.

REVIEWERS' COMMENTS:

Reviewer #1 (Remarks to the Author):

I reviewed the paper entitled “targeting the transmembrane cytokine co-receptor neuropilin-1 improves renal injury and renal fibrosis”.

There is a major problem with the data presented in this manuscript as NRP1 function is evaluated in renal tubular cells, a cell population where this co-receptor is not expressed.

The antibody used in this paper is certainly not specific to NRP1.

I have attached two files to support this statement.

In that sense, data presented in figures F, L and K are not correct.

NRP1 is expressed throughout the renal vasculature: glomeruli and peritubular capillaries, this is well known and documented. For this reason, the authors should have questioned their findings; the renal interstitium should be highly positive (endothelial cells, pericytes) and so should be the glomeruli (negative glomerulus shown in human biopsies in Figure 1L, bottom).

The authors then carried out a tubular conditional knock out of Nrp1 using the Ksp-ice (tubular specific) and again, it is very striking (and surprising) that they observed a downregulation of NRP1/Nrp1 in their knock out using WB and q-PCR. Statistical significance $**P < 0.01$ (impossible considering the lack of expression of NRP1 in renal tubular cells). The co-IP presented in Figure 1I is also questionable, considering the fact that we don't know which antibody was used.

I am afraid that NRP1 signalling is certainly not abrogated during fibrosis progression in their experimental settings as it is highly active in fibroblasts/myofibroblasts and endothelial cells. Data presented in figure 1B (together with the supportive data I have included) show that NRP1 signalling is active in fibroblasts/myofibroblasts and ECs during fibrosis

progression.

NRP1 promotes fibrosis progression in the liver (doi: 10.1172/JCI41203) , cell autonomously, in hepatic stellate cells, which are interstitial vascular pericyte-like cell that transdifferentiate into myofibroblasts (deposition of extracellular matrix, etc..).

In that sense, it is very likely that NRP1 signalling in renal interstitial cells is a key component of fibrosis progression in the kidney.

Reviewer #2 (Remarks to the Author):

1. The roles of Neuropilin-1 in both AKI and CKD were studied by other groups along with different mechanisms. In this research work, the authors also did the research work on the role of Neuropilin-1 in AKI-induced CKD, which increased the understanding on Neuropilin-1 in kidney diseases to some degree. Compared to the existing reports, the authors should better stress the novelty of this research work on Neuropilin-1 in kidney diseases.
2. A lot of omics analyses data were shown to support the conclusion. However, the specific quantitative data showing the protein levels of fibrosis, tubular injuries, etc, based on western blotting and ELISA should be performed and provided.
3. Too much observational data from omics were used to support the conclusion without the interventions between the components of the mechanism raised by the authors. It is known that many phenomenons could be observed by omics analyses which could be the secondary responses. Thus, intervention works are needed to support the research conclusion of this study in logic.

Reviewer #3 (Remarks to the Author):

We know that repeated instances of acute kidney injury (AKI) can lead to chronic kidney disease (CKD). And a common characteristic of most cases of CKD is renal fibrosis. So a deeper understanding of the renal fibrogenesis pathway is important in identifying new therapies for AKI/CKD. We do know that TGF- β signaling is a key driver of organ fibrosis, including in the kidney, but therapies targeting TGF- β have so far proven to be non-efficacious in the clinic, as well as showing deleterious on- and off-target effects. This suggests that other parts of the renal fibrosis pathway are playing an important role in the

process.

We know that neuropilin-1 (NRP1) is a broad-spectrum cytokine co-receptor, including for TGF- β and TNFs (it's also a co-receptor for the spike protein of SARS-CoV-2 so it's profile recently has risen dramatically). But its role in organ fibrosis is underexplored. So the authors decided to explore it here in AKI in mice using a ischemia-reperfusion model (the most commonly used AKI model), as well as in patients with renal transplant-associated kidney insufficiency.

In this study, the authors elucidated the potential mechanisms underlying kidney injury and fibrosis development by focusing on NRP1 and TGF- β . They explored in detail the role and mechanism of NRP1 in renal injury and fibrosis. They further validated NRP1 is upregulated in distal renal tubular cells of patients with transplanted kidney dysfunction and renal ischemia-reperfusion injury mice, and knockout of NRP1 in mice reduced the levels of renal injury and fibrosis. Mechanistically, NRP1 positive DT cells secrete collagen and communicate with myofibroblasts, and activate Smad3 to exacerbate renal fibrosis. Meanwhile, they also found the increased TNF- α after renal injury promote binding of NRP1 to TNF- α receptor. The binding of receptors affects the downregulation of lysine crotonylation of metabolic enzyme, leading to a decrease in cellular oxidative phosphorylation levels and exacerbating kidney damage. Furthermore, they validated that the dual blockade of NRP1 and TGFR1 leads to an improvement in renal injury and fibrosis compared to inhibiting either one alone. Overall, the authors have presented a substantial volume of data from animal and in vitro studies to substantiate their conclusion.

The paper is well-written, and the article introduces novel perspectives, presenting a comprehensive narrative. However, there are several suggestions for enhancing the manuscript, as outlined below:

- 1) I understand that knocking out NRP1 exhibits a protective effect in the 30-minute ischemia-reperfusion model. However, does NRP1 confer the same protective effect when kidney damage is more severe?

2) The authors proposed that NRP1 is closely related to various cell death modes through single cell transcriptomic data, but only primarily verified its impact on promoting apoptosis. It is recommended to further validate the involvement of other forms of programmed cell death mentioned in the article through immunostaining or other methods. This would contribute to a more comprehensive conclusion in the article.

3) In Figure 6H, the font size and clarity of "PDGF β " and "Etv6" in the nucleus should be consistent.

4) The representation of the time points for ischemia-reperfusion injury in the manuscript is not entirely consistent. For example, in some places, it is written as "IR D5," while in others, it is written as "IRD5." This should be standardized.

5) In Extended data Fig4A, the "Nrp1" enclosed within the dashed box represents gene names and should be italicized.

6) In Extended data Figure 4D, the font for "Relative correlation" corresponding to "Myo" and its associated numerical values should be in bold, consistent with the legend "Nrp1+DT" on the left.

7) I've noticed that in some figure legends, the authors have indicated the number of animals used in each group, as seen legends of Extended Data Figure 5, while in other figure legends, such as legends of Extended Data Figure 6, this information is not provided, even though it can be inferred directly from the images.

Overall, the manuscript is well-organized and written with substantial work. Thus, I would recommend its acceptance with minor revisions.

Editorial Note: Reviewer #1 attachment #1

Nrp1 visualization by scRNAseq in various mouse models of kidney fibrosis progression

First model: Fibrosis progression after acute kidney injury (IR, ischemia reperfusion) in the mouse, from *Kirita et al, 2020, PNAS* DOI: [10.1073/pnas.2005477117](https://doi.org/10.1073/pnas.2005477117)

[figure redacted]

Nrp1 is highly expressed Fib/Per (Fibroblast/pericytes) and detected in Endothelial Cells
Note the absence of Nrp1 expression in distal tubules (red bracketed region)

Second model: Diabetic Kidney Disease from *Wu et al, 2022, Cell metabolism*
<https://doi.org/10.1016/j.cmet.2022.05.010>

Diabetic Kidney Disease (DKD) is characterised by a high level of interstitial fibrosis and glomerulosclerosis. The mouse model which mimics human progressive DKD is used in this paper. **db/m is the control DKD group** receiving no treatment.

[figure redacted]

Nrp1 is highly expressed in EC (endothelial cells) and Fib (Fibroblast/myofibroblasts)
Note the absence of Nrp1 expression in distal tubules (red bracketed region)

Third model: UUO (unilateral ureteral obstruction) and IR (ischemia reperfusion) in the mouse *Li et al, 2022, Cell metabolism*
DOI: [10.1016/j.cmet.2022.09.026](https://doi.org/10.1016/j.cmet.2022.09.026)

[figure redacted]

Nrp1 is highly expressed in Fib (Fibroblast/myofibroblasts) and EC (endothelial cells)
Note the absence of Nrp1 expression in distal tubules (red bracketed region)

Nrp1 visualization by IF in various mouse models of kidney fibrosis progression (our data, unpublished): Folic acid and UUO

No expression of NRP1 in renal tubules

[figure redacted]

Nrp1 visualization by IF after Acute Kidney Injury (our published work, DOI:<https://doi.org/10.1016/j.kint.2021.12.028>)

[figure redacted]

Editorial note: Reviewer #1 attachment #2

**Nrp1 visualization by IF in wt mouse: our work below (unpublished)
which corroborates data from:**

DOI: [10.1016/j.kint.2016.10.010](https://doi.org/10.1016/j.kint.2016.10.010)

And

DOI: [10.1152/ajprenal.00311.2017](https://doi.org/10.1152/ajprenal.00311.2017)

No expression of NRP1 in renal tubules

[figure redacted]

Preliminary Rebuttal:

Reviewer #1

Reviewer #1 (Remarks to the Author):

1.I reviewed the paper entitled “targeting the transmembrane cytokine co-receptor neuropilin-1 improves renal injury and renal fibrosis”.

There is a major problem with the data presented in this manuscript as NRP1 function is evaluated in renal tubular cells, a cell population where this co-receptor is not expressed.

Response:

This is a very important issue, so we have made extensive revisions to the previous draft to address it, including the following modifications:

- 1) Firstly, we obtained authorization from Dr. Benjamin Humphreys to use the public data in his website (<http://humphreyslab.com/SingleCell/>) (Figure 1C in the revised manuscript). Based on this spatial transcriptomic data ^[1], *Nrp1* expression in distal renal tubular cells framed by black lines is higher after ischemia-reperfusion (I-R). The specific authorization explanation is on line 827-829, the image description is on line 120-121, and the figure legend is on line 1022-1023 in the revised manuscript (clean version), respectively.
- 2) Secondly, we obtained single-nuclei data from mice with nephrotic syndrome ^[2] and COVID-19 autopsy donors ^[3] through publicly available database websites at <https://singlecell.broadinstitute.org/> (Extended Figure 2A-2B in the revised manuscript). From the images directly obtained from the website, it can be seen that *Nrp1* has a high expression level in the distal renal tubules with red fonts. The specific image description is on line 124-128, and figure legend is on line 1131-1134 in the revised manuscript (clean version), respectively.
- 3) We also downloaded single-cell data from different mouse kidney injury models of GSE197266 ^[4] for analysis, and found that *Nrp1* expression is higher in the distal renal tubules after folic acid (FA), sodium oxalate (SO), ischemia reperfusion injury (IRI) and unilateral ureteral obstruction (UUO) (Extended Figure 2C in the revised manuscript). The specific image description is on line 124-128, and figure legend is on line 1134-1136 in the revised manuscript (clean version), respectively.

- 4) We also obtained spatial transcriptome data of healthy kidneys from <https://www.spatialomics.org/SpatialDB/>^[5] (Extended Figure 2D in the revised manuscript). From the figure, it can be seen that *Nrp1* and *Cdh16* are co-expressed in the same cells. As *Cdh16* is a general maker for distal tubules, these findings further support the notion that *Nrp1* is expressed in distal renal tubular cells. The specific image description is on line 124-128, and figure legend is on line 1136-1137 in the revised manuscript (clean version), respectively.
- 5) We also further confirmed the expression of *Nrp1* in renal tubular epithelial cells using antibodies from different companies, such as Santa Cruz, R&D and Abcam (Extended Data Figure 2E in the revised manuscript). The specific antibody information is on line 548-552, image description is on line 128-130, and figure legend is on line 1137-1139 in the revised manuscript (clean version), respectively.
- 6) Next, we used FISH (Fluorescence in Situ Hybridization) technology and found that the *Nrp1* probe co-stains with *Cdh16* protein expression in the same cells. The specific sequence information of the probe is shown in Table S2. The fluorescent image is in

Figure 1D in the revised manuscript. The specific experimental method is on line 557-571, image descriptions is on line 121-124, and figure legend is on line 1023-1024 in the revised manuscript (clean version), respectively.

Table S2. Sequence for Nrp1 Used for Fluorescence in Situ Hybridization (Related to methods section)

Gene	Sequence(5'to3')
Nrp1	GTAACCGGGAGATGTGAGGTACCC
	AGGATTCGAGTCTTGCTCCAGGTC
	TGGATAGAACGCCTGAAGAGGAGC
	TGTGGCTCTCTCAGGGTAGATCCT
	CCAGAAGGTCATACAGTGGGCAGA

- 7) In addition, we have replaced the immunohistochemical images of renal transplant patients with immunofluorescent images (Figure 1N). In immunofluorescence images, except for renal tubules, the staining of glomeruli is more pronounced compared to immunohistochemical images. The specific image description is on line 162-165, and figure legend is on line 1032 in the revised manuscript (clean version), respectively.
- 8) We also added immunofluorescence staining comparison images before and after knocking out renal tubular Nrp1 in Extended Figure 3C in the revised manuscript to verify the knockout effect of Nrp1. From the figure, it can be seen that Nrp1 expression is reduced in renal tubules, and the knockout does not affect the expression of Nrp1 in endothelial cells. The specific image description is on line 184-185, and figure legend is on line 1145 in the revised manuscript (clean version), respectively.

- 9) We added immunofluorescence staining comparison images of myofibroblasts that are knocked out for Nrp1 compared to controls in Extended Figure 10D in the revised manuscript to verify the knockout effect of Nrp1. From the figure, it can be seen that Nrp1 expression is reduced in the renal interstitium, and the knockout does not affect the expression of Nrp1 in blood vessels. The specific image description is on line 373-375, and figure legend is on line 1233-1234 in the revised manuscript (clean version), respectively.

2. The antibody used in this paper is certainly not specific to NRP1.

Response:

We have added specific antibody information to the manuscript. In the initial manuscript, we used antibodies from Abcam company (Abcam, UK, ab81321). To verify the expression of Nrp1, we also used antibodies from Santa Cruz Biotechnology (Santa Cruz Biotechnology, USA, sc-5307) and R&D Systems (R&D Systems, USA, FAB5994P). All these antibodies demonstrated similar distribution patterns of NRP1 in renal tubules, indicating that the antibody used in this paper is specific to NRP1. The specific antibody information is on line 548-552 in the revised manuscript (clean version), respectively.

3. I have attached two files to support this statement.

In that sense, data presented in figures F, L and K are not correct.

Response:

We appreciate the additional effort to support the reviewer's claim. But below are the images from the papers/unpublished data supplied:

Editorial note: the following figures are all redacted

First model: Fibrosis progression after acute kidney injury (IR, ischemia reperfusion) in the mouse, from *Kirita et al, 2020, PNAS* DOI: [10.1073/pnas.2005477117](https://doi.org/10.1073/pnas.2005477117)

Second model: Diabetic Kidney Disease from *Wu et al, 2022, Cell metabolism*
<https://doi.org/10.1016/j.cmet.2022.05.010>

Diabetic Kidney Disease (DKD) is characterised by a high level of interstitial fibrosis and glomerulosclerosis. The mouse model which mimics human progressive DKD is used in this paper. **db/m is the control DKD group** receiving no treatment.

Third model: UUO (unilateral ureteral obstruction) and IR (ischemia reperfusion) in the mouse *Li et al, 2022, Cell metabolism*
DOI: [10.1016/j.cmet.2022.09.026](https://doi.org/10.1016/j.cmet.2022.09.026)

Nrp1 visualization by IF in various mouse models of kidney fibrosis progression (our data, unpublished): Folic acid and UUO

No expression of NRP1 in renal tubules

Folic Acid (FA) induced fibrogenesis: Nrp1 is expressed in the renal interstitial space during fibrosis progression

Nrp1 visualization by IF after Acute Kidney Injury (our published work, DOI:<https://doi.org/10.1016/j.kint.2021.12.028>)

Nrp1 visualization by IF in wt mouse: our work below (unpublished) which corroborates data from:

DOI: [10.1016/j.kint.2016.10.010](https://doi.org/10.1016/j.kint.2016.10.010)

And

DOI: [10.1152/ajprenal.00311.2017](https://doi.org/10.1152/ajprenal.00311.2017)

No expression of NRP1 in renal tubules

After carefully studying the files provided, we have the following thoughts:

While we acknowledge the expression of NRP1 in renal endothelial cells and fibroblasts, there is indeed expression in renal tubular epithelial cells, as well. Firstly, regarding the reference to the absence of Nrp1 expression in the snRNA-seq (Fig 1), in our analysis using the same data (from *Kirita et al, 2020, PNAS DOI: 10.1073/pnas.2005477117*)^[6] the loop of Henle (LOH) was included in the distal renal tubules, and in fact in this figure the average expression of Nrp1 increased from 0 to 1.5-fold in MTAL and CTAL2 (estimated from the iconography) as shown in the right of Fig 1 (yellow bracketed region). We believe that due to the very high level of expression of Nrp1 in pericytes and endothelial cells, its relative expression in distal tubes appears non-existent but this is not the case – there is indeed expression in this region of the kidney.

Meanwhile, Wu *et al.* mentioned in their study published in JASN^[7]: “single-nucleus RNA sequencing (snRNA-seq) offers comparable gene expression quantitation (despite reduced mRNA in the nucleus compared with the whole cell) as well as substantial advantages over single-cell RNA sequencing (scRNA-seq). These include representation of rare or fragile kidney cell types, the ability to use archival frozen samples, elimination of dissociation-induced transcriptional stress responses, and successful performance on

[figure redacted]

inflamed fibrotic kidney.” But when they “compared unsupervised clustering results using a matched set of epithelial cell and nucleus transcriptomes”, they found that “snRNA-seq enriches for a smaller proportion of different genes than scRNA-seq”, further reporting that “we could identify only four clusters from the snRNA-seq dataset, whereas the scRNA-seq dataset yielded seven clusters” in tubular cells. Therefore, we understand that snRNA-seq has advantages in detecting glomerular cells and a broader range of interstitial cells, but scRNA-seq has advantages in detecting epithelial cells. Hence, the partial discrepancies between the single-nucleus RNA sequencing results depicted in Fig 1 and our single-cell sequencing results are to be expected.

Meanwhile, we discovered in other data published in JASN by the same corresponding author^[1] that Nrp1 expression was elevated in TAL, DTL and PC (yellow bracketed region), as part of distal renal tubules, in an ischemia-reperfusion model (IR) (Fig. 2), resembling our scRNA-seq results. After obtaining authorization from Dr. Benjamin Humphreys, this data has been placed in Figure 1C of our revised manuscript, as mentioned in our above response. Consequently, we inferred an increased Nrp1 expression in the distal tubules during I-R.

Nrp1 expression is increased after IR in TAL , DTL and PC in distal tubes

Dixon, E. E., et al. (2022). "Spatially Resolved Transcriptomic Analysis of Acute Kidney Injury in a Female Murine Model." *Journal of the American Society of Nephrology* 33(2): 279-289. doi: 10.1681/ASN.2021081150. Epub 2021 Dec . doi: 10.1681/ASN.2021081150. Epub 2021 Dec 1.

[figure redacted]

As with your indication of Nrp1 expression in endothelial cells, we acknowledge its expression in these cells. Our research was supported by the grant provided by the National Natural Science Foundation of China (Grants No.81974087), titled "Endothelial progenitor cells regulated angiogenesis and inhibited renal interstitial fibrosis by mediating neuropilin-1 secreted from exosomes". Initially, our intention was to observe the impact of targeting Nrp1 in endothelial cells on the progression of kidney disease. However, based on the aforementioned data, we observed a decrease in Nrp1 expression in endothelial cells following I-R, while there was an increase in Nrp1 expression in the distal renal tubules. Our single-cell and fluorescence data also confirmed the elevated Nrp1 expression in the distal renal tubules. Considering studies suggesting Nrp1's potential role in promoting fibrosis^[8-12] and the critical involvement of distal renal tubules in kidney diseases, we hypothesized that augmenting Nrp1 specifically in renal tubules might aid in suppressing kidney fibrosis. Consequently, we shifted our research direction.

In response to your reference to a second model of diabetic kidney disease (DKD), we

[figure redacted]

believe that the problem is again that you attribute the data to the exclusion of LOH cells and the fact that Nrp1 in the distal tubules is just not relatively as high. In fact, we observed in another dataset^[13] published in NCOMMS by the same corresponding author that Nrp1 expression was increased after diabetic kidney disease (DKD) in ATL in distal tubes in the right of Fig 3 (yellow bracketed region). In fact, Nrp1 expression was also increased in proximal tubule (PT) (Fig. 3).

For further validation, we made further observations by analyzing the scRNA-seq data published in PNAS^[14] and found that Nrp1 was expressed in both DCT/CT (yellow bracketed region) and PCT (green bracketed region), rather than not in renal tubules as your stated (Fig. 4).

Nrp1 was expressed after DKD in DCT/CT

Fig4

Wilson, P. C., et al. (2019). "The single-cell transcriptomic landscape of early human diabetic nephropathy." *Proceedings of the National Academy of Sciences of the United States of America* 116(39): 19619-19625 doi: 10.1073/pnas.1908706116. Epub 2019 Sep 10.

[figure redacted]

Regarding the unilateral ureteral obstruction (UJO) and the I-R models you mentioned, we believe that DTL-ATL, which you boxed, and TAL, which you did not box, do express Nrp1 (Fig. 5). Furthermore, we observed an increased expression of Nrp1 in the Failed Repair PT within the data presented in that article.

[figure redacted]

Regarding the immunofluorescence staining you mentioned, the antibody we used was from Abcam (ab81321, lot number GR212288-48). In fact, we did co-staining of NRP1 and vascular-specific marker CD31 at the beginning of the study and found that NRP1 was expressed in glomeruli and co-expressed with CD31, but we did not elaborate further because it was not related to the topic of our manuscript. We also observed Nrp1 expression in the renal tubules (as shown below in Fig. 6).

[figure redacted]

In our revised manuscript, we also added immunofluorescence staining comparison images before and after knocking out renal tubular Nrp1 in Extended Figure 3C (in revised manuscript). From the figure, it can be seen that Nrp1 is expressed in endothelial cells. The specific image description is on line 184-185, and figure legend is on line 1145 in the revised manuscript (clean version), respectively.

Regarding your mention of co-staining of Nrp1 with α -SMA in the UJO model, we also utilized the UJO model and performed co-staining of Nrp1 with α -SMA and found that they co-stain, but we also found expression of Nrp1 in renal tubular epithelial cells (Fig.7).

[figure redacted]

Regarding Nrp1 expression in the folic acid (FA) model, we have not utilized this model, so we cannot elaborate on the fluorescent expression of Nrp1 in it, but when we used different mouse kidney injury models of GSE197266 for analysis, we found that *Nrp1* expression increased in the distal renal tubules after folic acid induced renal injury in Extended Figure 2C in the revised manuscript.

In addition, as we observed the expression of Nrp1 on pericytes and myofibroblasts, we constructed pericyte-specific and myofibroblast-specific Nrp1 gene knockdown mice, respectively, and found that renal fibrosis was reduced after knocking down Nrp1 in pericytes or myofibroblasts. This section was included in the revised manuscript as Extended Data Figure 9 and Extended Data Figure 10.

Overall, we find your concern regarding Nrp1 expression in renal tubules to be unfounded. We do acknowledge the presence of NRP1 in glomeruli and the renal interstitium, as we have also observed this phenomenon. However, we maintain our assertion that renal tubular cells do express Nrp1. We believe that your study aligns with ours and doesn't conflict with our findings.

4.NRP1 is expressed throughout the renal vasculature: glomeruli and peritubular capillaries, this is well known and documented. For this reason, the authors should have questioned their findings; the renal interstitium should be highly positive (endothelial cells, pericytes) and so should be the glomeruli (negative glomerulus shown in human biopsies in Figure 1L, bottom).

Response:

As we also found that NRP1 is expressed throughout the renal vasculature, including glomeruli and peritubular capillaries, we agree with your suggestions and incorporate the recommended changes into the manuscript.

Firstly, we have replaced the immunohistochemical images of renal transplant patients with immunofluorescent images (Figure 1N). In immunofluorescence images, except for renal tubules, the staining of glomeruli is more pronounced compared to immunohistochemical images. The specific image description is on line 162-165, and figure legend is on line 1032 in the revised manuscript (clean version), respectively

Secondly, we also added immunofluorescence staining comparison images before and after knocking out renal tubular Nrp1 in Extended Figure 3C in the revised manuscript. From the figure, it can be seen that Nrp1 expression is reduced in renal tubules, and the knockout does not affect the expression of endothelial cells. The specific image description is on line 184-185, and figure legend is on line 1145 in the revised manuscript (clean version), respectively.

5.The authors then carried out a tubular conditional knock out of Nrp1 using the Ksp-ice (tubular specific) and again, it is very striking (and surprising) that they observed a downregulation of NRP1/Nrp1 is their knock out using WB and q-PCR. Statistical significance $P<0.01$ (impossible considering the lack of expression of NRP1 in renal tubular cells). The co-IP presented in Figure 1I is also questionable, considering the fact that we don't know which antibody was used.**

Response:

We sincerely thank you for careful reading. Firstly, as we mentioned earlier, we believe that Nrp1 is expressed in renal tubular epithelial cells, so a decrease in Nrp1 expression after knocking out Nrp1 in renal tubular epithelial cells is certainly reasonable.

Secondly, we apologize once again for not disclosing the specific information about the antibodies. We have added the specific information in the revised manuscript. In the initial manuscript, we used antibodies from Abcam (Abcam, UK, ab81321). To verify the expression of Nrp1, we also used antibodies from Santa Cruz Biotechnology (Santa Cruz Biotechnology, USA, sc-5307) and R&D Systems (R&D Systems, USA, FAB5994P) in the revised manuscript. The specific antibody information is on line 548-552 in the revised manuscript (clean version), respectively.

6. I am afraid that NRP1 signalling is certainly not abrogated during fibrosis progression in their experimental settings as it is highly active in fibroblasts/myofibroblasts and endothelial cells. Data presented in figure 1B (together with the supportive data I have included) show that NRP1 signalling is active in fibroblasts/myofibroblasts and Ecs during fibrosis progression.

Response:

Since we have also found that NRP1 is expressed in renal myofibroblasts, we agree with your view that it is highly active in fibroblasts/myofibroblasts and endothelial cells. Thus, we specifically knocked out *Nrp1* in myofibroblasts by using *Nrp1^{fl/+},Col1a2-icre* mice, and we supplemented relevant immunofluorescence images (Extended Figure 10D in the revised manuscript) and found that specific knockout of *Nrp1* in renal myofibroblasts also resulted in less kidney fibrosis. These results suggest that targeting NRP1 has a broad-spectrum anti-renal fibrosis effect, which is not limited to renal tubules, but does not conflict with the conclusion of this paper that renal tubule NRP1 is involved in renal fibrosis. From the figure, it can be seen that *Nrp1* expression is expressed in renal tubules and is reduced in the renal interstitium, and the knockout does not affect the expression of *Nrp1* in blood vessels. The specific image description is on line 373-375, and figure legend is on line 1233-1234 in the revised manuscript (clean version), respectively.

7. NRP1 promotes fibrosis progression in the liver (doi: 10.1172/JCI41203), cell autonomously, in hepatic stellate cells, which are interstitial vascular pericyte-like cell that transdifferentiate into myofibroblasts (deposition of extracellular matrix, etc..).

In that sense, it is very likely that NRP1 signalling in renal interstitial cells is a key component of fibrosis progression in the kidney.

Response:

We agree with your view that it is likely that NRP1 signaling in renal interstitial cells is a key component of fibrosis progression in the kidney. In fact, we alleviated renal fibrosis by knocking out *Nrp1* from pericytes or myofibroblasts in the renal interstitium, which further confirms this viewpoint.

However, at the same time, studies have found that renal tubular epithelial cells are closely related to fibrosis [15-17]. We found that distal renal tubular cells secrete collagen and communicate with myofibroblasts. In addition, there is a large proportion of renal tubular epithelial cells present in the kidneys, so we believe that *Nrp1* also plays a pro-fibrotic role in distal renal tubular epithelial cells.

Reviewer #2

Reviewer #2 (Remarks to the Author):

1. The roles of Neuropilin-1 in both AKI and CKD were studied by other groups along with different mechanisms. In this research work, the authors also did the research work on the role of Neuropilin-1 in AKI-induced CKD, which increased the understanding on Neuropilin-1 in kidney diseases to some degree. Compared to the existing reports, the authors should better stress the novelty of this research work on Neuropilin-1 in kidney diseases.

Response:

Thank you for your constructive comments. We modified the Discussion section to better point out the novelty of the study and its findings.

In this study, we comprehensively investigated the role of Nrp1 in both AKI and CKD. Firstly, we confirmed for that Nrp1 is expressed in DT cells, and its expression is upregulated following I-R. Knockdown of Nrp1 in DT cells improved I-R-induced renal injury and subsequent renal fibrosis. Secondly, we demonstrated that Nrp1 interacts with the TNF- α receptor Tnr1a, influencing downstream signaling pathways, ultimately leading to decreased Kcr of Cox4i1 in renal tubular epithelial cells. This inhibition of aerobic metabolism in renal tubular epithelial cells led to cell death. Finally, we innovatively discovered that *Nrp1*+ DT cells possess pro-fibrotic properties by secreting collagen. Meanwhile, we further confirmed that the communication between *Nrp1*+ DT cells and myofibroblasts establishes a vicious cycle of mutual activation through reciprocal secretion of pro-fibrotic factors, accelerating renal fibrosis.

The specific description is on line 463-472 in the revised manuscript (clean version).

2. A lot of omics analyses data were shown to support the conclusion. However, the specific quantitative data showing the protein levels of fibrosis, tubular injuries, etc, based on western blotting and ELISA should be performed and provided.

Response:

We appreciate the importance of WB and ELISA for quantitative protein analysis for fibrosis and tubular injuries. We supplemented some Western blotting and ELISA data as follows:

Firstly, we supplemented the data of mouse plasma Kim1 through ELISA detection, with Figure 2D corresponding to day 5 after I-R, and Figure 2E corresponding to day 14 after I-R. The specific experimental method is on line 588-595. The Figure 2D image description is on line 194-195, and figure legend is on line 1040-1047 in the revised manuscript (clean version), respectively. The Figure 2E image description is on line 197-198, and figure legend is on line 1047-1049 in the revised manuscript (clean version), respectively.

In addition, we added western blotting analysis in Figure 3B to confirm that the expression of Nfkb1, Smad3, and fibrosis-related markers (α -SMA and Pdgfrb) are higher after I-R, while knocking out Nrp1 reduced their expression. The specific antibody information is on line 576-580. The image description is on line 218-221, and figure legend is on line 1057-1059 in the revised manuscript (clean version), respectively.

3. Too much observational data from omics were used to support the conclusion without the interventions between the components of the mechanism raised by the authors. It is known that many phenomenons could be observed by omics analyses which could be the secondary responses. Thus, intervention works are needed to support the research conclusion of this study in logic.

Response: Given the complexity of our manuscript, it's challenging to intervene in every aspect. However, we believe performing intervention experiments on key and pivotal components of the mechanisms outlined in the paper are both feasible and crucial.

In fact, we have conducted some intervention experiments, such as demonstrating significant improvement in renal injury by simultaneously inhibiting the expression of Nrp1 and Tgfrb1 by injecting siRNA into the renal capsule of mice (Extended Data Figure 10A-10B). The image description is on line 367-369, and figure legend is on line 1222-1229 in the revised manuscript (clean version), respectively.

At the same time, we added Tnr1a or Tgfr1siRNA to primary renal tubular epithelial cells, demonstrating that Nrp1 interacts with Tnr1a and Tgfr1 to activate downstream Nfkb1 and Smad3 pathways, respectively (Figure 6C). The image description is on line 369-373, and figure legend is on line 1111-1112 in the revised manuscript (clean version), respectively.

In addition, we transfected the mutant plasmid into primary renal tubular epithelial cells, demonstrating that the Cox4i1 K29 from lysine to arginine resulted in a decrease in detectable levels of lysine crotonylation modification and a decrease in mitochondrial count, indicating that the lysine residue at the K29 site of Cox4i1 is crucial for maintaining mitochondrial function (Figure 4C-4D). The image description is on line 282-285, and figure legend is on line 1079-1081 in the revised manuscript (clean version), respectively.

Regarding the mechanism, we also confirmed that binding of Nfkb1 to the promoter region of *Etv6* in NRP1-positive cells through CUT&RUN experiments (Figure 3E). The image description is on line 227-231, and figure legend is on line 1061-1062 in the revised manuscript (clean version), respectively.

Reviewer #3

Reviewer #3 (Remarks to the Author):

We know that repeated instances of acute kidney injury (AKI) can lead to chronic kidney disease (CKD). And a common characteristic of most cases of CKD is renal fibrosis. So a deeper understanding of the renal fibrogenesis pathway is important in identifying new therapies for AKI/CKD. We do know that TGF- β signaling is a key driver of organ fibrosis, including in the kidney, but therapies targeting TGF- β have so far proven to be non-efficacious in the clinic, as well as showing deleterious on- and off-target effects. This suggests that other parts of the renal fibrosis pathway are playing an important role in the process.

We know that neuropilin-1 (NRP1) is a broad-spectrum cytokine co-receptor, including for TGF- β and TNFs (it's also a co-receptor for the spike protein of SARS-CoV-2 so it's profile recently has risen dramatically). But its role in organ fibrosis is underexplored. So the authors decided to explore it here in AKI in mice using a ischemia-reperfusion model (the most commonly used AKI model), as well as in patients with renal transplant-associated kidney insufficiency.

In this study, the authors elucidated the potential mechanisms underlying kidney injury and fibrosis development by focusing on NRP1 and TGF- β . They explored in detail the role and mechanism of NRP1 in renal injury and fibrosis. They further validated NRP1 is upregulated in distal renal tubular cells of patients with transplanted kidney dysfunction and renal ischemia-reperfusion injury mice, and knockout of NRP1 in mice reduced the levels of renal injury and fibrosis. Mechanistically, NRP1 positive DT cells secrete collagen and communicate with myofibroblasts, and activate Smad3 to exacerbate renal fibrosis. Meanwhile, they also found the increased TNF- α after renal injury promote binding of NRP1 to TNF- α receptor. The binding of receptors affects the downregulation of lysine crotonylation of metabolic enzyme, leading to a decrease in cellular oxidative phosphorylation levels and exacerbating kidney damage. Furthermore, they validated that the dual blockade of NRP1 and TGFR1 leads to an improvement in renal injury and fibrosis compared to inhibiting either one alone. Overall, the authors have presented a substantial volume of data from animal and in vitro studies to substantiate their conclusion.

The paper is well-written, and the article introduces novel perspectives, presenting a comprehensive narrative. However, there are several suggestions for enhancing the manuscript, as outlined below:

1.1 understand that knocking out NRP1 exhibits a protective effect in the 30-minute ischemia-reperfusion model. However, does NRP1 confer the same protective effect when kidney damage is more severe?

Response:

We express our heartfelt gratitude for your invaluable feedback, which has significantly contributed to enhancing the quality of our manuscript. We established severe I-R models with ischemia durations of 45 minutes followed by reperfusion for 5 and 45 days, respectively. Our experiments indicate that, consistent with the 30-minute I-R model, knockdown of Nrp1 in DT cells mitigates renal injury and fibrosis when more severe kidney damage occurs. We have organized the data into figures; however, due to the abundance of supplementary images, we have not yet integrated this data into the manuscript. If you and the editor deem it necessary to include them, we will reconsider the layout of the figures and incorporate them into the manuscript later.

2. The authors proposed that NRP1 is closely related to various cell death modes through single cell transcriptomic data, but only primarily verified its impact on promoting apoptosis. It is recommended to further validate the involvement of other forms of programmed cell death mentioned in the article through immunostaining or other methods. This would contribute to a more comprehensive conclusion in the article.

Response:

We appreciate this valuable suggestion. However, due to space limitations, we have only focused on exploring the relationship between Nrp1 and apoptosis, and the relationship between NRP1 and other forms of cell death will be the focus of further mechanistic research in our future articles.

3. In Figure 6H, the font size and clarity of "PDGFβ" and "Etv6" in the nucleus should be consistent.

Response:

We have made modifications to ensure the font size and clarity of "PDGFβ" and "Etv6" in the nucleus is consistent.

4. The representation of the time points for ischemia-reperfusion injury in the manuscript is not entirely consistent. For example, in some places, it is written as "IR D5," while in others, it is written as "IRD5." This should be standardized.

Response:

We sincerely thank you for careful reading. Based on your suggestion, we have corrected all "IR D1" to "IRD1", "IR D5" to "IRD5", "IR D14" to "IRD14", "UUO D14" to "UUOD14", and "5/6Nx D14" to "5/6NxD14" in the images and manuscript.

5. In Extended data Fig4A, the "Nrp1" enclosed within the dashed box represents gene names and should be italicized.

Response:

Thanks for your careful reading. Due to adjustments made to the layout of our figures and in the interest of streamlining supplementary images, the previous Extended Data Figure 4A has been removed. However, we believe your suggestion remains insightful, and we have reviewed both the images and the manuscript to ensure that similar errors are not repeated. Thanks for your correction.

6. In Extended data Figure 4D, the font for "Relative correlation" corresponding to "Myo" and its associated numerical values should be in bold, consistent with the legend "Nrp1+DT" on the left.

Response:

Thank you for helping us correct this mistake. The previous Extended Data Figure 4D has been relocated to Figure 5E. We have made modifications to Figure 5E based on your suggestions.

7. I've noticed that in some figure legends, the authors have indicated the number of animals used in each group, as seen legends of Extended Data Figure 5, while in other figure legends, such as legends of Extended Data Figure 6, this information is not provided, even though it can be inferred directly from the images.

Response:

In our revised manuscript, this oversight has been corrected. We have carefully checked the figure legends of all statistical images to ensure that they clearly labeled the number of samples contained in each group. Specifically, we have added the sample size for each group in the figure legends of Figure 3B, 4G, 5F, and Extended Data Figures 3E, 3I-3K, 4A-4E, 6C, 7, and 10A-10C, and 10E, respectively.

8. Overall, the manuscript is well-organized and written with substantial work. Thus, I would recommend its acceptance with minor revisions.

Response: We thank you for your support of our study and for your valuable comments, which have helped improve the clarity and strength of our paper.

- [1] DIXON E E, WU H, MUTO Y, et al. Spatially Resolved Transcriptomic Analysis of Acute Kidney Injury in a Female Murine Model [J]. *Journal of the American Society of Nephrology*, 2022, 33(2): 279-289.
- [2] SIDHOM E H, KIM C, KOST-ALIMOVA M, et al. Targeting a Braf/Mapk pathway rescues podocyte lipid peroxidation in CoQ-deficiency kidney disease [J]. *Journal of Clinical Investigation*, 2021, 131(5).
- [3] DELOREY T M, ZIEGLER C G K, HEIMBERG G, et al. A single-cell and spatial atlas of autopsy tissues reveals pathology and cellular targets of SARS-CoV-2 [J]. *bioRxiv*, 2021.
- [4] CHEN Z, LI Y, YUAN Y, et al. Single-cell sequencing reveals homogeneity and heterogeneity of the cytopathological mechanisms in different etiology-induced AKI [J]. *Cell Death & Disease*, 2023, 14(5): 318.
- [5] FAN Z, CHEN R, CHEN X. SpatialDB: a database for spatially resolved transcriptomes [J]. *Nucleic Acids Research*, 2020, 48(D1): D233-D237.
- [6] KIRITA Y, WU H, UCHIMURA K, et al. Cell profiling of mouse acute kidney injury reveals conserved cellular responses to injury [J]. *Proceedings of the National Academy of Sciences of the United States of America*, 2020, 117(27): 15874-15883.
- [7] WU H, KIRITA Y, DONNELLY E L, et al. Advantages of Single-Nucleus over Single-Cell RNA Sequencing of Adult Kidney: Rare Cell Types and Novel Cell States Revealed in Fibrosis [J]. *Journal of the American Society of Nephrology*, 2019, 30(1): 23-32.
- [8] CAO S, YAQOOB U, DAS A, et al. Neuropilin-1 promotes cirrhosis of the rodent and human liver by enhancing PDGF/TGF-beta signaling in hepatic stellate cells [J]. *Journal of Clinical Investigation*, 2010, 120(7): 2379-94.
- [9] BALL S G, BAYLEY C, SHUTTLEWORTH C A, et al. Neuropilin-1 regulates platelet-derived growth factor receptor signalling in mesenchymal stem cells [J]. *Biochemical Journal*, 2010, 427(1): 29-40.
- [10] ARAB J P, CABRERA D, SEHRAWAT T S, et al. Hepatic stellate cell activation promotes alcohol-induced steatohepatitis through Igfbp3 and SerpinA12 [J]. *Journal of Hepatology*, 2020, 73(1): 149-160.
- [11] ABDULLAH A, AKHAND S S, PAEZ J S P, et al. Epigenetic targeting of neuropilin-1 prevents bypass signaling in drug-resistant breast cancer [J]. *Oncogene*, 2021, 40(2): 322-333.
- [12] ZHANG J, QIU J, ZHOU W, et al. Neuropilin-1 mediates lung tissue-specific control of ILC2 function in type 2 immunity [J]. *Nature Immunology*, 2022, 23(2): 237-250.
- [13] WILSON P C, MUTO Y, WU H, et al. Multimodal single cell sequencing implicates chromatin accessibility and genetic background in diabetic kidney disease progression [J]. *Nat Commun*, 2022, 13(1): 5253.
- [14] WILSON P C, WU H, KIRITA Y, et al. The single-cell transcriptomic landscape of early human diabetic nephropathy [J]. *Proceedings of the National Academy of Sciences of the United States of America*, 2019, 116(39): 19619-19625.
- [15] LIVINGSTON M J, SHU S, FAN Y, et al. Tubular cells produce FGF2 via autophagy

after acute kidney injury leading to fibroblast activation and renal fibrosis [J]. *Autophagy*, 2023, 19(1): 256-277.

- [16] XU L, SHARKEY D, CANTLEY L G. Tubular GM-CSF Promotes Late MCP-1/CCR2-Mediated Fibrosis and Inflammation after Ischemia/Reperfusion Injury [J]. *Journal of the American Society of Nephrology*, 2019, 30(10): 1825-1840.
- [17] LUO C, ZHOU S, ZHOU Z, et al. Wnt9a Promotes Renal Fibrosis by Accelerating Cellular Senescence in Tubular Epithelial Cells [J]. *Journal of the American Society of Nephrology*, 2018, 29(4): 1238-1256.

REVIEWERS' COMMENTS

Reviewer #1 (Remarks to the Author):

Review Nature Comm NRP1 fibrosis

The initial concern regarding NRP1 upregulation in damaged tubular cells has been well addressed and we are now convinced this cell compartment is indeed NRP1 positive upon injury, although this is not the major site of de novo production. The single cell data and the new IF on wt and ko are both convincing so I have no further comment regarding that point.

We also appreciated the effort to validate omics observations at functional levels for key mechanistic studies.

However, we think the chronological aspect of AKI, AKI to CKD transition, maladaptive repair and CKD progression could be better defined. What we learnt here is that Nrp1 plays a key role in AKI to CKD transition by promoting tubular cell death and driving fibrosis, thereby precipitating CKD progression. 5 days post IR corresponds to the repair phase and this is where AKI to CKD transition is initiated so this is correct.

But we don't know the role of tubular NRP1 in AKI, in the acute phase, in the first 24 to 48 hours following the hit, this is not addressed (Nrp1 upregulation is seen as early as 4 hours post IR and could be indeed implicated in the acute phase).

So it is important to rephrase some parts of the paper, and replace tubular NRP1's novelty in what's already known as driver of AKI to CKD progression (role of endothelium, inflammation, hypertrophy/polyploidy and FiB/pericytes).

What we want to stress here is that difference in kidney function and CKD progression at later stages could be directly linked to how nrp1 ko handled the initial acute phase. The greater the damage is to the kidney during the acute phase, the worse the outcome to CKD progression will be at later phases. Do nrp1 ko and wt handle AKI the same way? Similar kidney damage at 24/48hrs post IR? similar kidney function at 24/48 hrs post IR? Level of tubular injury at 24/48 hrs post IR? we are not saying that more experiments should be

performed but if the authors have these data, we think this should be mentioned/added.

If both wt and nrp1ko handle the initial acute phase equally, the role of tubular nrp1 in the repair phase and during CKD transition will be more striking.

Reviewer #2 (Remarks to the Author):

All my concerns have been addressed. I have no more any comments.

Reviewer #3 (Remarks to the Author):

The author of this study has made revisions based on review comments and meets the publication standards of Nature Communication. In general, the author modified some of the writing problems in the original version, also replied and supplemented our questions, and made the logic clearer by modifying the layout of some pictures. It is recommended to accept it directly.

Point-by-point response to the reviewers' comments

Reviewer #1

1. We think the chronological aspect of AKI, AKI to CKD transition, maladaptive repair and CKD progression could be better defined.

Response:

We believe your opinion is very constructive. Many articles use 24 hours after IR to construct AKI models. On the 5th day after IR, it is more of an AKI to CKD transition process. Therefore, we have revised the section in the manuscript that describes 5 days post IR as AKI. Specifically, as follows:

- 1) The text in lines 100-101 "Knockout of Nrp1 in renal DT epithelial cells ameliorates AKI and subsequent chronic renal fibrosis." has been changed to "Knockout of Nrp1 in renal DT epithelial cells ameliorates kidney injury and subsequent chronic renal fibrosis."
- 2) The text in lines 211-212 " In summary, these findings demonstrate that Nrp1 in distal tubules promotes AKI and subsequent chronic kidney fibrosis." has been changed to "In summary, these findings demonstrate that Nrp1 in distal tubules promotes kidney injury and subsequent chronic kidney fibrosis."
- 3) The text in lines 397-399 "Based on *in vivo* and *in vitro* studies, we conclude that Nrp1 exacerbates acute kidney injury by inhibiting aerobic metabolism and promotes CKD progression by regulating myofibroblast activation and collagen secretion." has been changed to "Based on *in vivo* and *in vitro* studies, we conclude that Nrp1 exacerbates kidney injury by inhibiting aerobic metabolism and promotes CKD progression by regulating myofibroblast activation and collagen secretion."

2. We don't know the role of tubular NRP1 in AKI, in the acute phase, in the first 24 to 48 hours following the hit, this is not addressed (Nrp1 upregulation is seen as early as 4 hours post IR and could be indeed implicated in the acute phase).

So it is important to rephrase some parts of the paper, and replace tubular NRP1's novelty in what's already known as driver of AKI to CKD progression (role of endothelium, inflammation, hypertrophy/polyploidy and FiB/pericytes).

Response:

In our study, we explored the effects of knocking out NRP1 in tubules one day after IR, and found that the results were consistent with those observed 5 days after IR. Since we observed the highest expression levels of NRP1 on the 5th day after IR, we chose to conduct subsequent studies on the 5th day post-IR. Additionally, this study suggests that NRP1 in the renal tubules promotes the progression of kidney disease by affecting aerobic metabolism during kidney injury, as discussed in the manuscript already. This does not fully align with some of the known drivers of AKI-CKD that you mentioned. However, we consider your suggestion to be very important. Thus, we have added these data in the revised manuscript in Supplementary Figure 3L-3M and corresponding Result section in

lines 207-209. We found NRP1 knockout in tubules one day after IR mitigated acute kidney injury and an improvement in renal function, accompanied by a decrease Kim1.

3. Do nrp1 ko and wt handle AKI the same way? Similar kidney damage at 24/48hrs post IR? similar kidney function at 24/48 hrs post IR? Level of tubular injury at 24/48 hrs post IR? we are not saying that more experiments should be performed but if the authors have these data, we think this should be mentioned/added.

If both wt and nrp1ko handle the initial acute phase equally, the role of tubular nrp1 in the repair phase and during CKD transition will be more striking.

Response:

These are good questions. Yes, nrp1 ko and wt handle AKI in the same way. The effects of NRP1 knockout in tubules one day after IR were explored. In comparison to the Vehi+IR group, the Tmx+IR group demonstrated a reduction in acute kidney injury and an improvement in renal function, accompanied by a decrease in the expression levels of renal tubular injury marker Kim1. Based on your suggestion, we have added some of these data in the revised manuscript in Supplementary Figure 3L-3M and corresponding Result section in lines 207-209, and added the following description at the end of the first paragraph of the Discussion section:

“Meanwhile, we explored the effects of knocking out NRP1 in tubules one day after IR, and found that the results were consistent with those observed 5 days after IR. Therefore, we believe that targeting the transmembrane cytokine co-receptor neuropilin-1 in distal tubules improves renal injury and fibrosis.”

Reviewer #2

All my concerns have been addressed. I have no more any comments.

Response:

Thank you very much for your time and support throughout the review process. We believe that the manuscript has been greatly improved based on your suggestions.

Reviewer #3

The author of this study has made revisions based on review comments and meets the publication standards of Nature Communication. In general, the author modified some of the writing problems in the original version, also replied and supplemented our questions, and made the logic clearer by modifying the layout of some pictures. It is recommended to accept it directly.

Response:

We are pleased to hear your positive feedback on our manuscript. We appreciate your recognition of our efforts to address the writing problems, respond to their questions, and improve the logical clarity by modifying the layout of some figures.